# Single-cell transcriptome atlas and chromatin accessibility landscape reveal differentiation trajectories in the rice root

Tian-Qi Zhang [1✉], Yu Chen[1,2], Ye Liu[3], Wen-Hui Lin [4] & Jia-Wei Wang [1,5✉]

Root development relies on the establishment of meristematic tissues that give rise to distinct cell types that differentiate across defined temporal and spatial gradients. Dissection of the developmental trajectories and the transcriptional networks that underlie them could aid understanding of the function of the root apical meristem in both dicots and monocots. Here, we present a single-cell RNA (scRNA) sequencing and chromatin accessibility survey of rice radicles. By temporal profiling of individual root tip cells we reconstruct continuous developmental trajectories of epidermal cells and ground tissues, and elucidate regulatory networks underlying cell fate determination in these cell lineages. We further identify characteristic processes, transcriptome profiles, and marker genes for these cell types and reveal conserved and divergent root developmental pathways between dicots and monocots. Finally, we demonstrate the potential of the platform for functional genetic studies by using spatiotemporal modeling to identify a rice root meristematic mutant from a cell-specific gene cohort.

[1] National Key Laboratory of Plant Molecular Genetics (NKLPMG), CAS Center for Excellence in Molecular Plant Sciences, Institute of Plant Physiology and Ecology (SIPPE), Chinese Academy of Sciences (CAS), Shanghai, China. [2] University of Chinese Academy of Sciences, Shanghai, China. [3] State Key Laboratory of Crop Genetics and Germplasm Enhancement, Key Laboratory of Landscaping, Ministry of Agriculture and Rural Affairs, College of Horticulture, Nanjing Agricultural University, Nanjing, China. [4] Joint International Research Laboratory of Metabolic & Developmental Sciences, School of Life Sciences & Biotechnology, Joint Center for Single Cell Biology, Shanghai Jiao Tong University, Shanghai, China. [5] ShanghaiTech University, Shanghai 200031, China. ✉email: tianqizhang@yeah.net; jwwang@sippe.ac.cn

Many dicotyledonous plants such as Arabidopsis have a root system composed of a single primary root and numerous lateral roots. By contrast, cereals such as *Oryza sativa* (rice) develop a dense fibrous root system consisting primarily of numerous postembryonic adventitious roots that develop from the stem (Supplementary Note 1)[1,2]. For both rice and Arabidopsis, root apical meristem (RAM) is formed during embryogenesis which generates the radicle or primary root after seed germination. The RAM is comprised of a stem cell niche (SCN) and undifferentiated small dividing cells in both dicots and monocots (Fig. 1b)[3,4]. In Arabidopsis, the SCN harbors quiescent center (QC) and surrounding stem cells which give rise to diverse root cell types including epidermis, root hairs, root cap, cortex, endodermis, and stele (pericycle, phloem, xylem, and pro-cambium) through cell division, expansion, and differentiation[5,6].

Rice is usually grown under flooding conditions and has developed distinct anatomy to adapt to this environment. Its radial root organization differs greatly from that of dicots such as Arabidopsis (Fig. 1b)[1,7]. The outer cell layers of rice roots are comprised of the epidermis, exodermis, and sclerenchyma. The exodermis (hypodermis with Casparian bands) of plant roots represents a barrier of variable resistance to the radial flow of both water and solutes[8,9]. Unlike other cereal crops, the rice root features a sclerenchyma layer which is hyper-suberized compared with the exodermis and consists of tightly packed small cells. The exodermis and sclerenchyma together form a barrier that reduces radial oxygen loss. The ground tissue of rice roots is composed of between 7 and 12 layers of cortical cells and one layer of endodermal cells (Fig. 1b)[2]. On the contrary, Arabidopsis roots only harbor one cortical and one endodermal cell layer. The increased number of rice cortical layers provide a venue for lysigenous aerenchyma formation and plays an essential role in adaptation to waterlogging conditions[10]. Taken together, although the anatomy of the rice root apical meristem has been intensively investigated, the molecular definition of its cell types is largely incomplete.

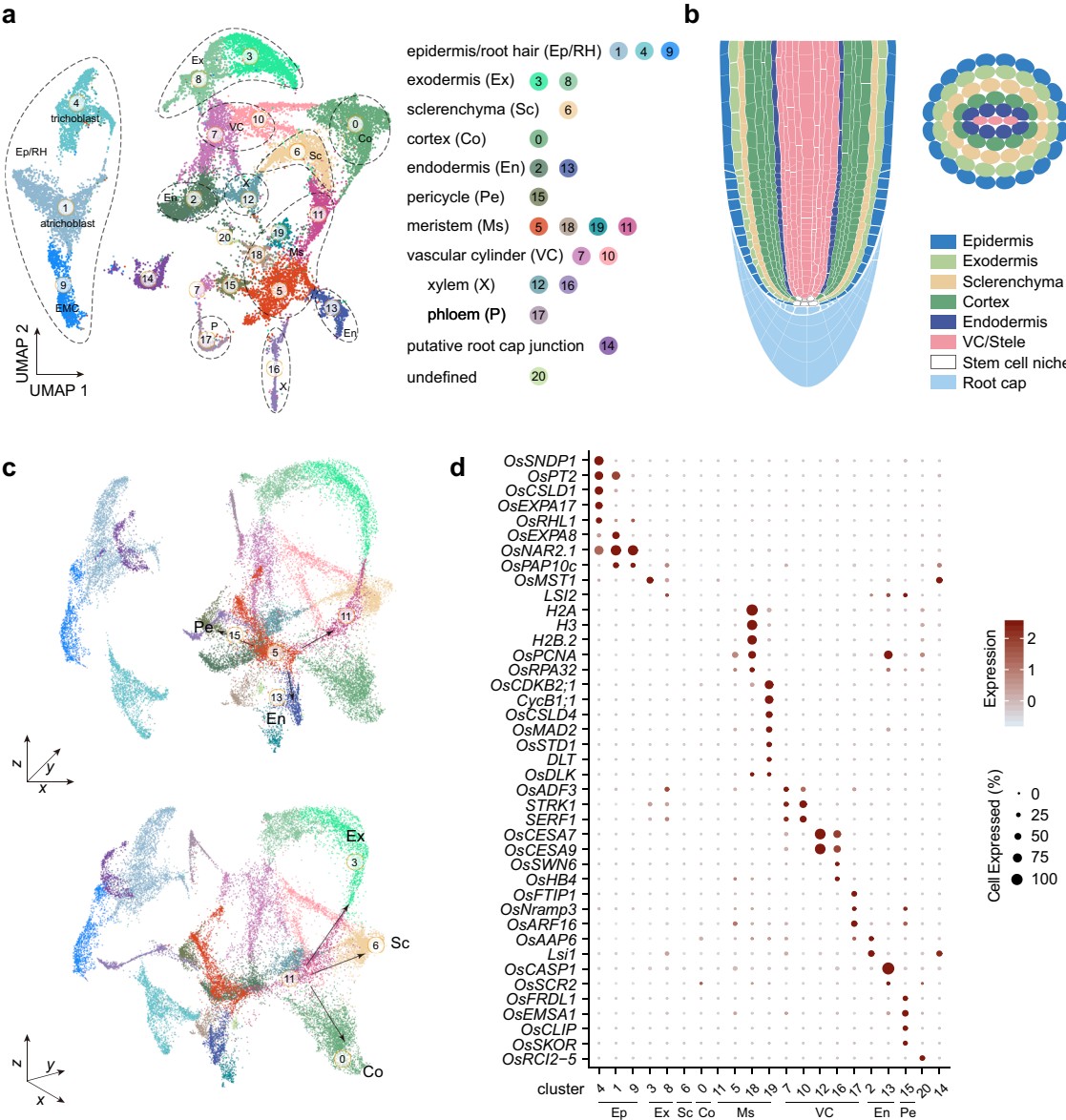

**Fig. 1 Generation of a rice radicle cell atlas. a** UMAP visualization of 21 cell clusters in rice radicles. Each dot denotes a single cell. Colors denote corresponding cell clusters. EMC, epidermal meristematic cell. **b** Schematic of anatomy of rice radicle. **c** Visualization of cell clusters by 3D UMAP scatterplots. Cluster names and colors are the same as in **a**. **d** Expression patterns of representative cluster-specific marker genes on UMAP. Dot diameter indicates the proportion of cluster cells expressing a given gene. The full names of selected genes are given in Supplementary Data 3.

Recent advances in single-cell RNA sequencing (scRNA-seq) technology provide unprecedented opportunities to systematically identify the entire cellular and molecular differentiation trajectory of plant stem cells at the single-cell level. Several scRNA-seq studies have revealed that Arabidopsis root tip cells have highly heterogeneous transcriptomes[11–18]. The transcriptome analysis of the individual cells at different developmental stages reveals a continuous differentiation trajectory of root development following their spatial distribution and temporal order[19–25].

## Results

**Generation of a rice radicle cell atlas**. To perform scRNA-seq of rice roots, the radicles of wild-type (ZH11) rice seedlings were harvested (~1.0 cm in length from root tip, $n = 90$), digested into protoplasts (plant cells without cell wall), and subjected to scRNA-seq assay using commercial 10× Chromium platform (10× Genomics, Supplementary Data 1). Two biological replicates were performed. The protoplasting and mitochondrial genes had little effects on the clustering (see Method; Supplementary Fig. S1a, c)[26]. The batch-effect between samples was removed by the Harmony algorithm (Supplementary Fig. S1b, e–g)[27]. We used standard computational pipelines (Cell Ranger provided by 10× Genomics) to align the raw sequencing data to the rice genome, and derived a gene expression matrix of 29,919 genes across 27,469 filtered cells.

After selecting highly variable genes, we used principal component analysis (PCA) on a gene expression matrix across 2000 variable genes and identified 100 statistically significant principal components (PCs) ($p < 0.05$). Using Seurat[28], these PCs were used to build a $k$-nearest neighbor graph of the cells, which was then partitioned into 21 transcriptionally distinct clusters. The analysis of cell cycle genes revealed that transcriptome heterogeneity was not dominated by cell cycle status (Supplementary Fig. S1h). To reveal local similarities and global structures of cell populations, the $t$-distributed stochastic neighborhood embedding ($t$-SNE) tool and uniform manifold approximation and projection (UMAP) algorithm were used (Fig. 1a; Supplementary Fig. S1d)[29,30]. Compared to the $t$-SNE, the UMAP offers faster runtime and consistency, meaningful organization of cell clusters, and preservation of continuums[29]. In contrast, the main advantage of the $t$-SNE is the ability to preserve local structure. Three-dimensional (3D) UMAP scatterplots were further constructed to determine the spatial and connective distribution of cell clusters (Fig. 1c; Supplementary Movie 1). By analyzing differentially expressed genes among the clusters, we identified a series of cluster-enriched genes and cluster-specific marker genes (Fig. 1d; Supplementary Data 2).

Because of the lack of specific markers for most cell types, we used the following two strategies to faithfully annotate cell clusters in the rice radicle cell atlas. First, we identified rice genes whose biological functions or expression patterns have been well studied (Supplementary Data 2; Supplementary Data 3; Supplementary Note 2 and 3). The examination of their expression pattern on the UMAP helped us to assign some cell clusters. Second, we performed RNA in situ hybridization assays for over 30 cluster-specific genes (see below, Supplementary Fig. S10). The expression pattern of these genes not only confirmed the above annotations, but also enabled us to assign unknown cell clusters. Overall, our results indicate that the rice root tip is composed of highly heterogeneous cells. The single-cell transcriptome atlas enabled us to identify most of the major cell types of the rice radicle (Supplementary Note 2). Clusters 1, 4 and 9 belonged to the epidermis/root hair population (Fig. 1a, see below). These clusters were separated from other cell clusters on the UMAP plot (Fig. 1a), suggesting a unique transcriptome signature. The

meristematic cell clusters (Clusters 5, 11, 18, and 19) localized in the center of the UMAP (Supplementary Note 2). Differentiation trajectories such as ground tissue (Clusters 0, 2, 3, 6, 8, and 13), pericycle (cluster 15) and vascular tissue (Clusters 7, 10, 12, 16, and 17) radiated away from the meristematic cell clusters (Fig. 1a, c). Notably, our gene expression and signature analyses provide fresh insights into cell type-specific physiological properties.

**The developmental trajectories of epidermal cells**. Since cells undergoing transition from one state to another (i.e., intermediate states) can be captured, scRNA-seq enables the exploration of the continuous differentiation trajectory of a developmental process[19–23]. Therefore, we first aimed to deduce the developmental trajectory of root epidermal cells. Re-clustering of three clusters belonging to the epidermis/root hair population (clusters 1, 4, and 9) revealed ten sub-cell clusters, named E0 to E9 (Fig. 2a). Interestingly, UMAP plotting revealed that cluster 9 topologically bifurcated into two trajectories, dominated by the cells in either cluster 1 or 4 (Fig. 2a). To validate this result, we conducted pseudotime analysis by ordering cells along a reconstructed trajectory using Palantir (Fig. 2b)[31]. Consistent with distribution distance on the UMAP, inferred trajectories demonstrated gradual transitions from cells in cluster 9 to early, mid, and late root-hair cells (clusters 4, trichoblasts) or to nonhair cells (clusters 1, atrichoblasts) (Fig. 2a; Supplementary Fig. S2a). Moreover, differentiation potential analysis indicated that the cells in clusters 1 and 4 were more differentiated than those in cluster 9 (Fig. 2c). Thus, these analyses collectively demonstrate that the cells in cluster 9 serve as epidermal meristematic cells. Along with cell division and differentiation, these cells progressively differentiate into nonhair cells (cluster 1) or root hair cells (cluster 4).

We chose Os10g0452700, Os03g0155500, Os01g0248900, and Os10g0578200 as representative genes for clusters 9, 1, and 4, respectively (Fig. 2d, e; Supplementary Fig. S2b). A survey of our scRNA-seq dataset revealed that the expression of Os10g0452700 gradually decreased along pseudotime, followed by increased transcription of Os03g0155500 (Fig. 2e). The expression of Os01g0248900 was elevated at the late developmental stage of atrichoblasts, while the progressive increase of Os10g0578200 expression was only observed in the trichoblast lineage (Fig. 2e).

The analyses of fluorescent (Venus-N7) reporters generally confirmed the expression patterns inferred by the scRNA-seq dataset. For example, expression of the Os10g0452700 reporter was mainly observed in epidermal cells of the root meristematic zone and decreased along with root development (Fig. 2f; Supplementary Fig. S2c). In contrast, the Os03g0155500 reporter only became active in meristematic cells that had started to differentiate (Fig. 2g). The Os01g0248900 and Os10g0578200 reporters showed complementary expression patterns (Fig. 2h, i): Os01g0248900 was strongly expressed in atrichoblasts, whereas the promoter of Os10g0578200 was exclusively activated in trichoblasts. Gene ontology (GO) analyses revealed further differences in the biological processes of atrichoblasts and trichoblasts: cluster 1 (atrichoblasts) was enriched for genes involved in cell wall biosynthesis, whereas cluster 4 (trichoblasts) showed gene signatures for fatty acid metabolic process (Supplementary Fig. S2d, e; Supplementary Data 4). Thus, the combination of scRNA-seq, pseudotime inference and reporter analyses enables us to reconstruct the progression of epidermal cell fate determination during atrichoblast and trichoblast development in rice.

**The developmental trajectories of ground tissues**. A major difference between monocots and dicots lies in the cell lineages of the ground tissues and epidermis. In monocots, the ground

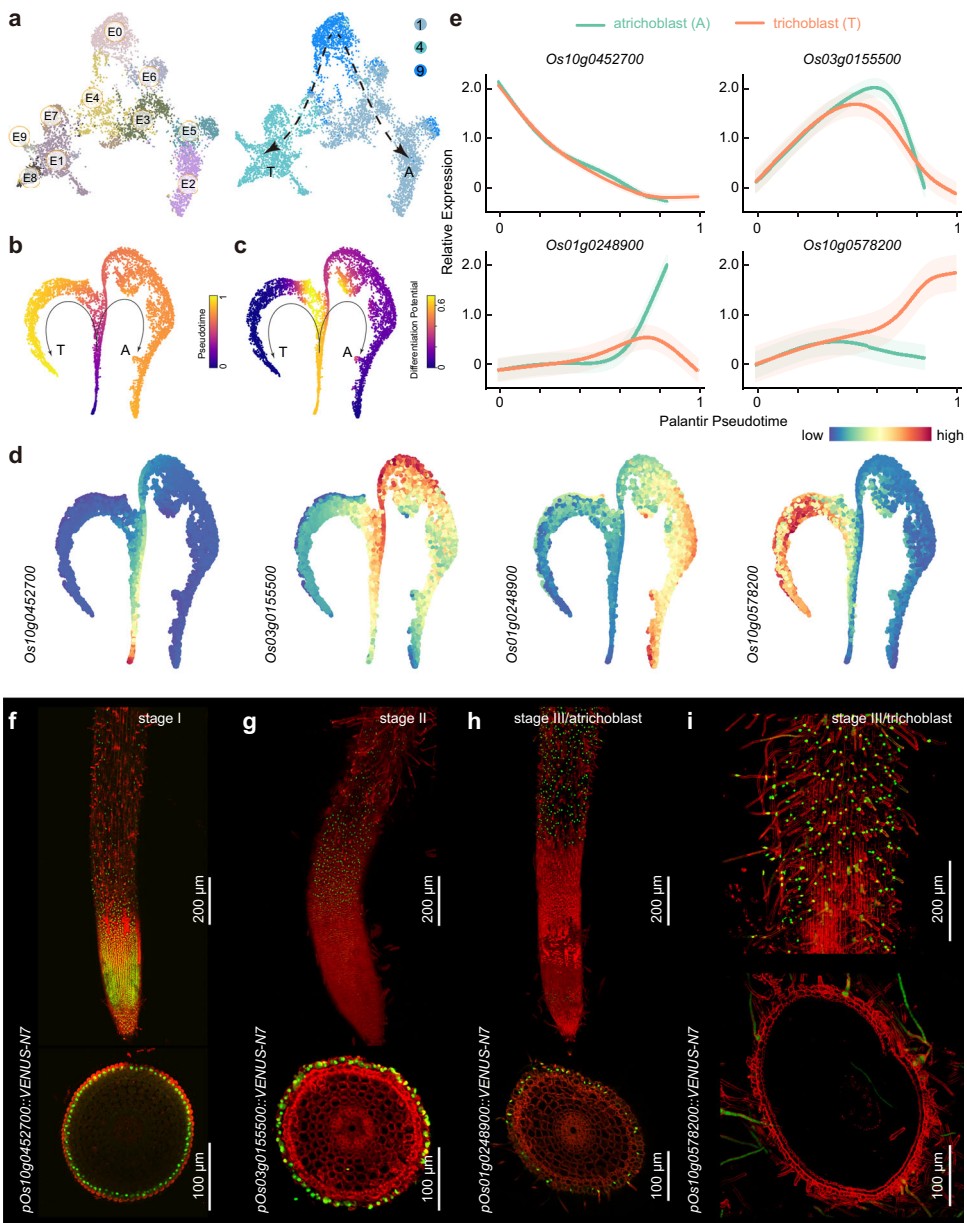

**Fig. 2 Differentiation trajectories of epidermal cells. a** UMAP projections showing epidermal cell populations (clusters 1, 4, and 9). E0 to E9, sub-cell clusters. Lines indicate two potential differentiation trajectories toward trichoblasts (T) and atrichoblasts (A). **b, c** t-SNE map of epidermal cell populations. Cells colored by Palantir pseudotime (**b**) or differentiation potential (**c**). **d** t-SNE map showing the expression pattern of four marker genes. **e** Expression of marker genes along Palantir pseudotime. **f–i** Expression of marker genes (green) in rice radicles. Three developmental stages (stage I to III) are shown. Six repeats each consisting of individual plants were performed for each gene. Red, FM4-64. Longitudinal (top) and transverse (bottom) sections are shown.

tissues and epidermis belong to a single common initial, while in dicots, the lateral root cap and epidermis share a common initial cell (Fig. 1b)[32]. The anticlinal division of the rice ground tissue/epidermis initial cell regenerates the initial cell and its daughter cell. The asymmetrical division of this daughter cell separates the epidermis and ground tissue lineages. Successive periclinal divisions of the ground tissue initial cell generate exodermis, sclerenchyma layers, cortex, and endodermis.

On the basis of functional annotations, clusters 3, 6, and 0 were designated as exodermis, sclerenchyma cell layer, and cortex, respectively (Fig. 1a). Intriguingly, closer examination of the distribution distances on the 3D UMAP and t-SNE plots revealed that all these clusters were connected to cluster 11 (Fig. 1a,c; Fig. 3a; Supplementary Fig. S3a), which was annotated as meristematic cells (Fig. 1a). Thus, this topology suggests that

cluster 11 may serve as common undifferentiated progenitor cells for root ground tissue (Fig. 3a; Supplementary Fig. S3a). In favor of this hypothesis, RNA velocity, ForceAtlas2 and differentiation potential analyses confirmed that the cells in cluster 11 reside in the "source" cell state, and those in clusters 3, 6, and 0 in end states (Fig. 3b–d; Supplementary Fig. S3b–e)[33,34].

Gene survey analysis uncovered several transcription factors highly expressed in cluster 11 (Fig. 3e). Expression of these genes, for example *OsGATA6* and *OsERF108* (*ETHYLENE RESPONSE FACTOR 108*), was progressively decreased along pseudotime (Fig. 3e, f). In line with this expression pattern, RNA in situ hybridization assay confirmed that *OsGATA6* and *GROWTH REGULATING FACTOR6* (*OsGRF6*) transcripts predominantly accumulated in the center of the RAM above the QC, and gradually decreased along the cell division zone until reaching the

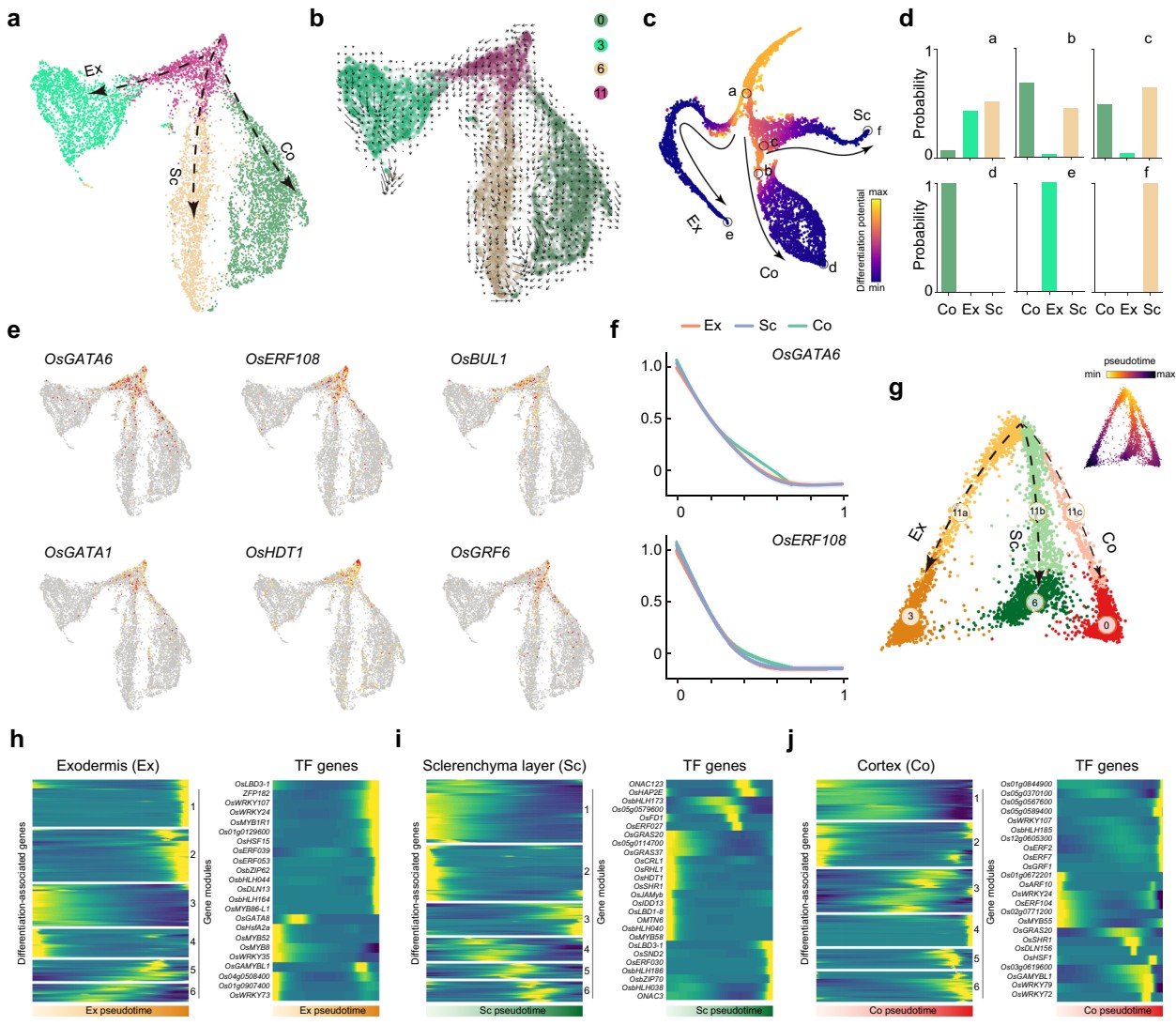

**Fig. 3 Differentiation trajectories of ground tissue. a** UMAP plot showing ground tissue cell populations. Ex, exodermis; Co, cortex; Sc, sclerenchyma layer. Cluster names (Clusters 0, 3, 6, and 11) and colors are the same as in Fig. 1a. **b** RNA velocity field projected onto the UMAP. Arrows represent average velocity and direction. Cluster names and colors are the same as in **a**. **c, d** t-SNE map of ground tissue cell populations. Cell colored by differentiation potential (**c**). Black lines mark three major differentiation trajectories toward Ex, Co, and Sc, respectively. a to f (black circles), six nodes used for quantification of branch probabilities (**d**). **e, f** Expression patterns of selected transcription factor genes on UMAP map (**e**) or along pseudotime (**f**). Expression of *OsGATA6* and *OsERF108* was greatly decreased along pseudotime during cell differentiation toward Ex, Co and Sc. **g** Pseudotime analysis of ground tissue differentiation trajectories by DPT. The cells in clusters 0, 3, 6 and 11 were grouped and re-clustered. Three differentiation trajectories are shown in different colors. Inset, pseudotime estimation. **h–j** Heatmap showing gene expression pattern during differentiation of Ex (**h**), Sc (**i**), and Co (**j**) along pseudotime (left column). Expression pattern of selected transcription factor (TF) along each differentiation trajectory is given (right column).

expansion and differentiation zone (Supplementary Fig. S4b, see below). Thus, the above results indicate that cluster 11 is composed of meristematic cells, giving rise to distinct cell types of ground and vascular tissues in the rice root tip.

To obtain a better understanding of the gene regulatory basis underlying ground tissue differentiation, we re-clustered the cells from clusters 0, 3, 6 and 11, and performed pseudotime and embedded heatmap analyses. We delineated three distinct cell lineages, leading to the formation of the exodermis, sclerenchyma cell layer and cortex (Fig. 3g). GO analyses indicated that the terminally differentiated cells in these lineages exhibited distinct gene signatures (Supplementary Fig. S3f–h; Supplementary Data 4). In addition, the transcription factors associated with their differentiation exhibited temporal expression patterns along pseudotime, and did not overlap (Fig. 3h–j). Taken together, our results reveal a differentiation continuum of meristematic cells

toward distinct ground tissues. Importantly, the inferred transcription factors provide a valuable resource for identification of key regulators for ground tissue differentiation.

The above analysis revealed that *OsGATA6* is expressed in the center of the RAM above the QC with gradated expression along the cell division gradient, suggesting that it may play an important role in the differentiation of ground and vascular tissue. To test this hypothesis, we generated *OsGATA6* knockout mutants using clustered regularly interspaced short palindromic repeats (CRISPR)-Cas9 technology (Supplementary Fig. S4a)[35–39]. The homozygous *Osgata6* mutants were dwarfed and developed short roots (Supplementary Fig. S4c). At the tissue level, the mutants had a shorter RAM, and exhibited defects in cell division and differentiation (Supplementary Fig. 4d). Thus, these findings substantiate the precision of scRNA-seq in identification of rice root mutants at the cell-specific level.

**Inference of transcriptional regulatory basis underlying cell differentiation by integrative analysis of ATAC-seq and scRNA-seq**. To further characterize the molecular basis for cell differentiation in the rice root tip, we performed assays for transposase-accessible chromatin sequencing (ATAC-seq), which probe changes in chromatin accessibility at a genome-wide level[40–45]. To this end, we harvested the root tissues from the meristematic zone (MZ) and elongation zone (EZ) (Fig. 4a). The comparison between the MZ and EZ samples revealed massive changes in chromatin accessibility (Fig. 4b, c; Supplementary Fig. S5a–c; Supplementary Data 5), suggesting that root meristematic cells undergo chromatin level reprogramming at chromatin level during cell differentiation.

We used the HOMER algorithm to characterize transcription factor binding motifs associated with variations in chromatin accessibility[46]. As shown in Fig. 4d, the MZ and EZ samples showed a distinct enrichment of transcription factor binding motifs and biological processes (Supplementary Fig. S5d; Supplementary Data 6). The binding motifs for transcription factor families such as bHLH, bZIP and GATA were highly enriched in the MZ, whereas MYB transcription factor binding sites were overrepresented in the EZ (Fig. 4d; Supplementary Fig. S5e). Consistent with these results, OsGATA6 (GATA transcription factor) and BRASSINOSTEROID UPREGULATED 1-LIKE1 (OsBUL1, a bHLH transcription factor), two genes highly expressed in cluster 11 (meristematic cells corresponding to exodermis, sclerenchyma cell layer, and cortex cells), exhibited open chromatin status only in the MZ (Fig. 4e). In contrast, the regulatory regions of two MYB transcription factor genes (cluster 12), MYB12 and MYB52, were closed in the MZ and became accessible in the EZ (Fig. 4e).

To test whether the chromatin accessibility of a given gene is correlated with its expression level during rice root development, we performed integrative analyses of scRNA-seq and ATAC-seq datasets. To this end, we assigned differentially accessible genes to the cell clusters revealed by our scRNA-seq experiment ("Methods"). The clusters belonging to meristematic cells (e.g., clusters 9 and 11) showed a high proportion of the genes with higher accessibility in the MZ (Fig. 4f, lower panel; Supplementary Fig. S6). On the contrary, cell clusters assigned to differentiated tissues (e.g., clusters 2, 7, 8, and 12) showed an enrichment of the genes that were accessible in the EZ (Fig. 4f, upper panel; Supplementary Fig. 6). For instance, the genes (Os10g0452700 and OsEXPA11) in cluster 9 (meristematic cells corresponding to epidermal cells) exhibited higher accessibility in the MZ than in the EZ (Fig. 4g; Supplementary Fig. S6j). In contrast, the genes (Os08g0112300, Os10g0490900, Os07g0176600, and OsPKS16) enriched in clusters 1 and 4 (corresponding to differentiated epidermal cells) were largely inaccessible in the MZ but became accessible in the EZ (Fig. 4g; Supplementary Fig. S6b, e). Notably, the changes in chromatin accessibility of these genes were correlated with their expression levels along pseudotime (Fig. 4g). Thus, the above results indicate that the analysis of chromatin accessibility across different developmental zones can reveal transcriptional regulatory mechanisms underlying cell differentiation at a cell-type-specific resolution.

**Inter-species comparison reveals conserved and divergent root developmental pathways**. Finally, because Arabidopsis root scRNA-seq datasets are already available, we wanted to explore conservation and divergence of the root cell types across the monocot-dicot divide. To this end, we first integrated three published Arabidopsis root scRNA-seq datasets[19,20,22], and re-performed cell clustering and assignment. In total, 22 cell clusters were identified (Supplementary Fig. S7). To compare gene expression across Arabidopsis and rice, we defined a set of one-to-one homologous genes ($n = 9,727$; Supplementary Data 7) across species using OrthoMCL[47]. Pairwise comparisons of root cell clusters of Arabidopsis and rice revealed a relatively high degree of similarities in the clusters corresponding to meristematic, epidermis (nonhair and root hair cells), phloem and xylem cells (Fig. 5a). Interestingly, the rice endodermis cluster showed a low correlation with the corresponding cluster from Arabidopsis but showed a positive correlation to Arabidopsis vascular tissue without phloem and xylem cells (VC$^{-P-X}$), suggesting these cell types are functionally aligned. The correlation between rice exodermis and Arabidopsis endodermis suggests that they are homologous tissues (Fig. 5a), consistent with their functionalities as paracellular transport barriers.

We next grouped Arabidopsis and rice scRNA-seq datasets and performed cell clustering analysis, which resulted in 30 super cell clusters (Fig. 5b; Supplementary Fig. S8). Since Arabidopsis and rice shared the highest one-to-one similarities for root hair, phloem, and xylem supercell clusters (Fig. 5b, c), we characterized them in greater details. For the root hair super cluster, gene clustering analyses revealed a list of core (conserved) genes that were highly expressed in root hairs in both species (Fig. 5d, right panel). Notably, some of these genes are known to play important regulatory roles in root hair development (Supplementary Data 9). For example, PRPL1 (PROLINE-RICH PROTEIN-LIKE 1) and EXPA7 (EXPANSIN A7), both of which were predominantly expressed in the root hair cluster (Fig. 5d, left panels), have been reported to control root hair elongation in Arabidopsis and rice[48,49]. Similarly, genes involved in root hair development such as RSH4 (ROOT HAIR SPECIFIC 4), CML25 (CALMODULIN LIKE 25), and RHD4 (ROOT HAIR DEFECTIVE 4) were also on the list of genes common to both species (Fig. 5d)[50–52]. The gene clustering analyses also identified species-specific programs (Fig. 5d, modules 3 and 4), suggesting unique properties of root hairs in each species. Similar to the observations for the root hair super cluster, we also successfully identified core gene lists corresponding to phloem and xylem cells (Fig. 5e, f). Taken together, the inter-species comparisons not only reveal evolutionally conserved genes involved in cell differentiation in the root tip, but also identify species-specific genes that may reflect fundamental differences in monocot/dicot root biology.

## Discussion

A fundamental problem in developmental biology is understanding how stem cells give rise to different cell types. To address this question, it is necessary to define the major cell types within a given tissue at both the anatomical and molecular levels. Compared to Arabidopsis, knowledge of rice root development is still limited. Although the anatomy of the rice root apical meristem has been intensively investigated, the molecular definition of its cell types is largely incomplete. Particularly, the inference of rice root cell types based on the expression of orthologous Arabidopsis genes is difficult because of variation in the number and nature of the cell types in monocots and dicots. For instance, the rice root contains exodermis and sclerenchyma cell layers, which are absent from Arabidopsis. Furthermore, the multilayered cortex of the rice root can differentiate into aerenchyma, an anatomical adaptation to water submergence, at late developmental stages, a capacity that is absent in Arabidopsis[53]. The expression map of the rice root tip at single-cell resolution presented in this study provides a valuable resource for defining different cell types. Based on our scRNA-seq and cell clustering, we successfully identified 21 cell types in the rice root tip. Importantly, in situ hybridization assays and promoter-reporter analysis revealed specific marker genes for most root tip cell types

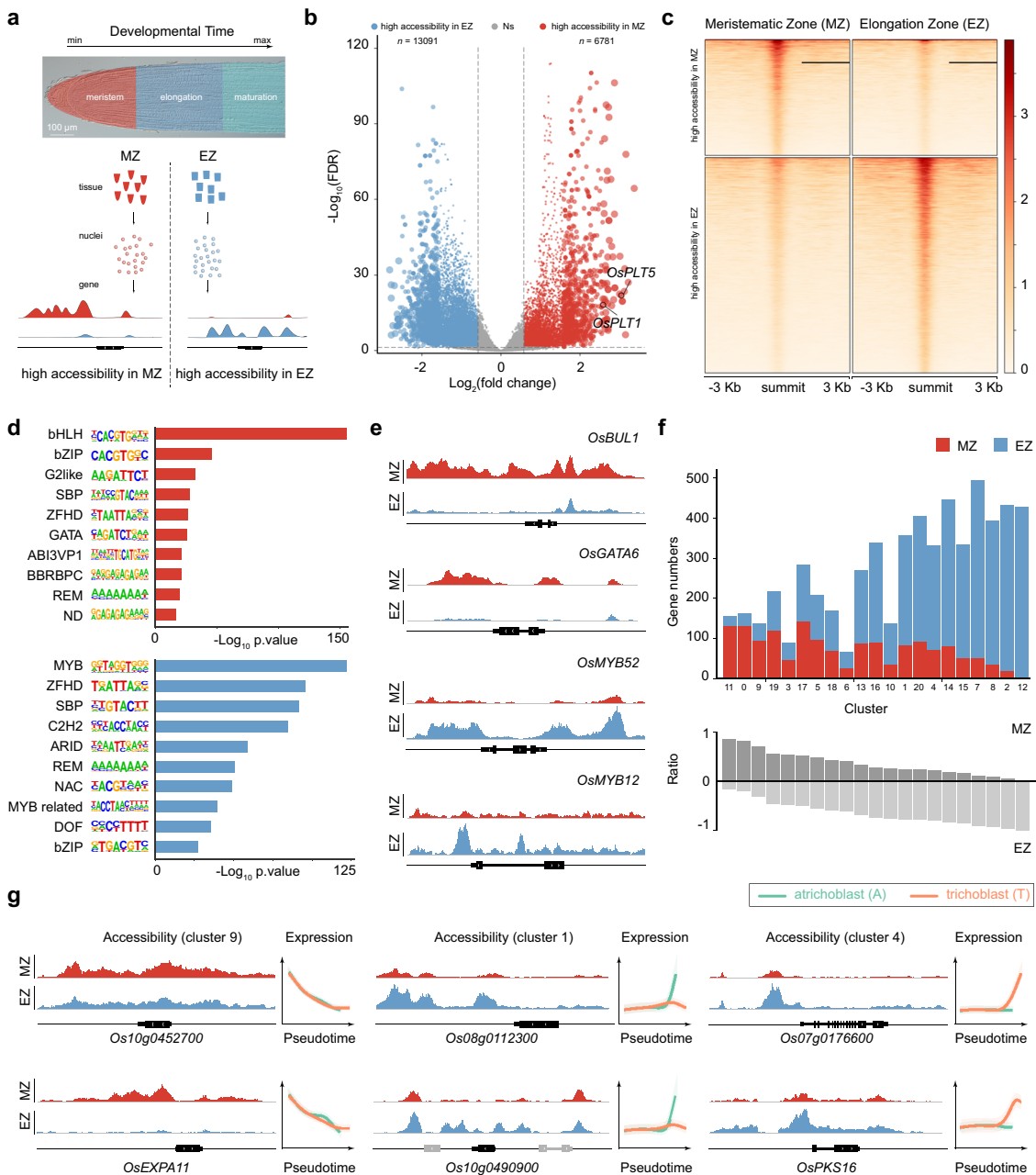

**Fig. 4 Inference of transcriptional regulatory basis by ATAC-seq and scRNA-seq. a** Schematic of tissue sampling. MZ, meristematic zone; EZ, elongation zone. **b** Volcano plot showing differentially accessible peaks between the MZ and EZ samples. Blue and red, highly accessible peaks in the EZ and MZ samples, respectively; Gray, no difference between the MZ and EZ samples. The number of differential peaks in each sample is given. FDR < 0.05; log$_2$(fold change) > 0.58 or <−0.58. **c** Pileup of ATAC-seq signals. Heat maps are ranked in decreasing order of ATAC-seq signal. Window size: peak summit ±3.0 kb. **d** Enrichment of transcription factor binding motifs within differential peaks in the MZ (red, top) and EZ (blue, bottom) samples. −Log$_{10}$(p value) for each binding motif is given. **e** Representative ATAC-seq tracks for bHLH (*OsBUL1*), GATA (*OsGATA6*) and MYB (*OsMYB12* and *OsMYB52*) transcription factor family genes. The genomic loci are shown, and the representative genes are highlighted in black. **f** Integrative analysis of scRNA-seq and ATAC-seq data. The genes associated with differential peaks identified in **c** were assigned to 21 cell clusters. The number of genes showing high accessibility in the MZ or EZ in each cell cluster is shown in different colors (top). The ratio (bottom) was calculated by the number of genes showing high accessibility in the MZ or EZ sample/total number of cluster-specific genes. **g** Representative ATAC-seq tracks for clusters 1, 4, 9 enriched genes. The genomic loci are shown, and the representative genes are highlighted in black. Two representative genes for each cluster are shown. Gene expression patterns during differentiation toward trichoblast (T) and atrichoblast (A) along the pseudotime are shown (right). Color annotation is the same in Fig. 2e.

(Supplementary Fig. S10). For example, we found that the transcripts of *Os04g0125700* are highly abundant in the root exodermis (Supplementary Fig. S10), whereas *Os01g0248900* and *Os10g0578200* are exclusively expressed in atrichoblasts and trichoblasts, respectively (Fig. 2h, i). Cell clusters corresponding to the cortex, exodermis and sclerenchyma layers (cluster 6) were also identified (Fig. 1a). Due to limitations imposed by sequencing depth and gene coverage, lowly expressed genes, which could include important transcription factors for cell fate determination, maybe not faithfully recovered in our scRNA-seq datasets. Nevertheless, the cell-type-specific reporters identified in this work can be used with cell-sorting methods to generate high

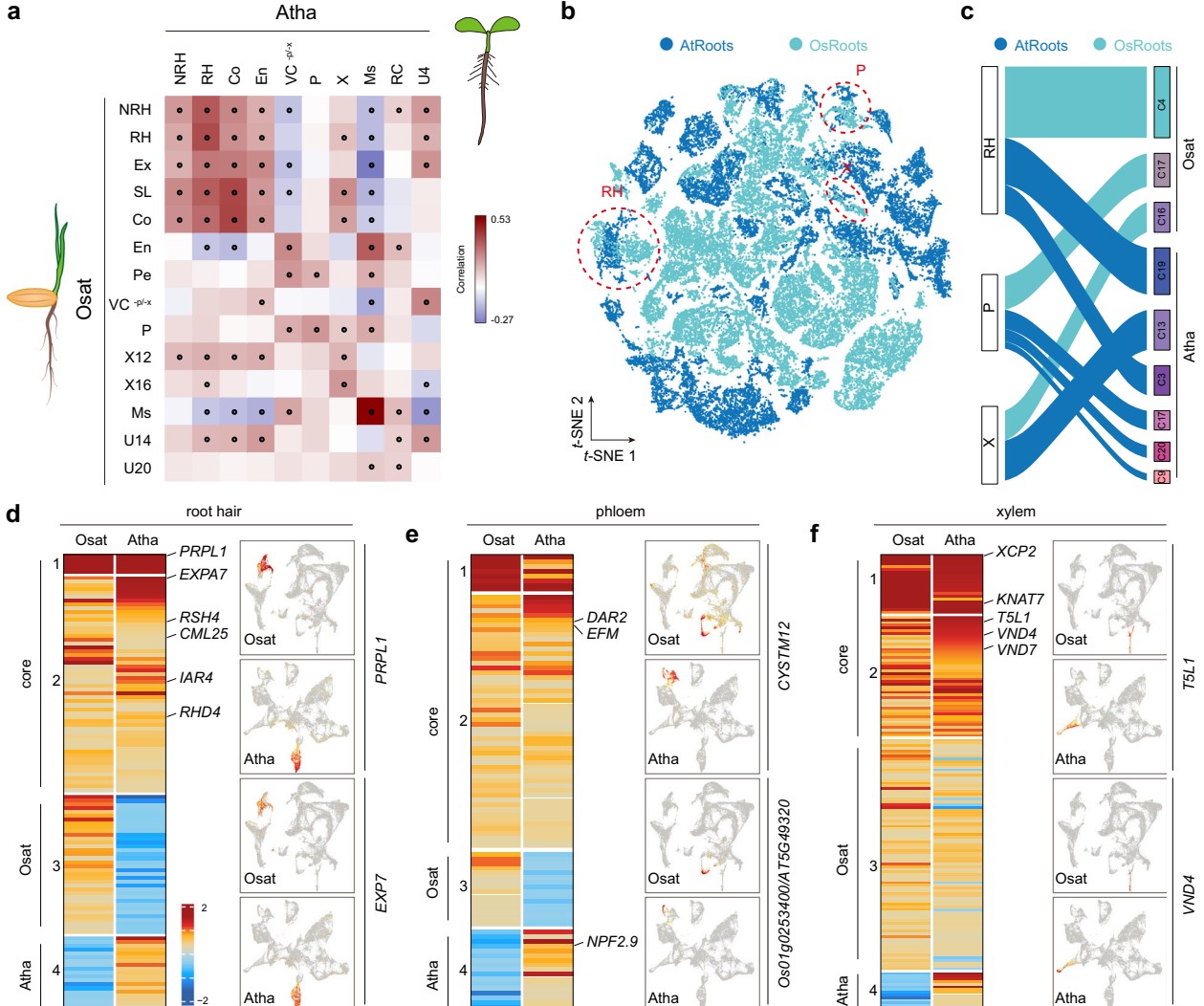

**Fig. 5 Comparison of rice and Arabidopsis root tips at single-cell resolution. a** Pairwise correlations of Arabidopsis (top) and rice (left) root cell clusters. Dots indicate statistically significant correlations. Atha, *A. thaliana*; Osat, *O. sativa*; NRH, non-root hair cells; RH, root hair; Ex, exodermis; SL, sclerenchyma layer; Co, cortex; En, endodermis; Pe, pericycle; VC$^{P-X}$, vascular tissue without phloem and xylem; P, phloem; X, xylem; X12 and X16, xylem clusters 12 and 16; Ms, meristematic cells; RC, root cap; U4, U14, and U20, unknown cell clusters 4, 14, and 20. **b** *t*-SNE plot showing 30 super cell clusters of Arabidopsis and rice root cells. Dotted circles mark the common RH, P, and X clusters, respectively. **c** Sankey diagram showing that Arabidopsis shares a high degree of similarities in RH, P, and X cell clusters with rice. Cluster number for Arabidopsis (Supplementary Fig. S7a) and rice (Fig. 1a) root cells is given on the right. **d-f** Gene clustering of RH (**d**), P (**e**), and X (**f**) clusters. For each cell type, core and species-specific sub-gene clusters (labeled with Atha or Osat) were identified. The known genes expressed in root hair, phloem and xylem are indicated. The expression pattern of selected core genes for each cell type is plotted on UMAP (right). The full names of selected genes are given in Supplementary Data 3.

quality cell-type-specific transcriptomes of the rice root tip, as has been done for Arabidopsis[54–56].

A major difference exists in the differentiation of ground tissue and root cap between dicots and monocots; the ground tissue and epidermis belong to a single common initial in rice, whereas lateral root cap and epidermis share a common initial cell in dicots such as Arabidopsis[32]. In monocots, the first asymmetrical division of the epidermis-endodermis initial cell produces two daughter cells, named as the epidermis cell initial and cortex-endodermis initial, with distinct lineages. The epidermis cell initially gives rise to the epidermis through successive periclinal divisions, whereas the cortex-endodermis initial cell subsequently generates the cortex and endodermis progenitor cells through asymmetric anticlinal divisions (Fig. 1c)[1]. How these distinct cell fates are specified in a step-wise manner from a common stem cell is largely unknown. Our results identified clusters 1/4/9, 3, 6, 0, and 2/13 as epidermis, exodermis, sclerenchyma cell layer,

cortex, and endodermis, respectively (Fig. 1a). Interestingly, pseudotime analysis revealed that the exodermis, sclerenchyma cell layer, and cortex are derived from a common cell cluster (cluster 11). Gene expression analyses indicates that cluster 11 is composed of mixed meristematic cells giving rise to distinct ground and vascular tissue cell types (Supplementary Fig. S4b), although we could not further separate them based on transcriptome differences. In the future, identification of marker genes for the cortex-endodermis initial cell and their use for generation of transgenic plants for cell fate mapping will help to precisely delineate the differentiation trajectories of specific ground tissue cell lineages[57]. Certainly, a similar approach can be used to elucidate how rice root cap cells differentiate from a single columella initial.

Forward genetics and quantitative trait loci mapping have identified a series of mutants showing defects in root meristem maintenance, crown root formation, root hair development, and

lateral root initiation[58–64]. Several functional orthologs of Arabidopsis root development mutants have also been studied in rice[65–68]. However, the number of available rice root mutants remains limited, preventing the accurate modeling of the gene regulatory network underlying rice root development. In this study, we have demonstrated the power of our scRNA-seq resource to accurately guide functional genetic studies of specific cell types. We show that mutation of *OsGATA6*, which is highly expressed in root meristematic cells (Supplementary Fig. S4), led to defects in cell division and differentiation. Therefore, we envision that exploration of our datasets will accelerate gene discovery and uncover more important regulators of rice root development or physiology in the near future.

Root architecture is a highly dynamic trait and contributes to plant adaptation to its environment. For example, drought tolerance in many plants, including Arabidopsis, rice, and maize, is associated with the ability generate a deep, wide-spreading, highly-branched root system[69–72]. The comprehensive definition of cell types by scRNA-seq lays the ground for understanding how root cell differentiation and root system architecture are shaped in response to environmental cues, such as drought, or flooding. Future work that generates roots scRNA-seq and ATAC-seq datasets of plants grown under different conditions or treatments should yield insights into how such environmental signals direct root growth.

## Methods

**Plant materials and growth conditions**. The wild-type *O. sativa* (ZH11) seedlings were used for root scRNA-seq experiments. Seeds were sowed on filter paper socked with water in petri dishes. After 5 days at 30 °C, the root tips of the seedlings were harvested for scRNA-seq experiment. For promoter Venus-N7 reporter lines, rice plants were grown in a growth chamber at 29 °C in long days (16 h light/8 h dark).

**Constructs**. The oligonucleotide primers are given in Supplementary Data 8. The promoter Venus-N7 reporters were generated by molecular cloning method. Briefly, the genomic DNAs of wild-type seedlings were extracted with EasyPure Plant Genomic DNA kit (Transgen, Cat No./ID: EE111-02). The regulatory fragments including promoter and 3' downstream sequence were PCR amplified and cloned into JW1696 using ClonExpress II one Step Cloning kit (Vazyme Biotech, Cat No./ID: C112). For RNA in situ hybridization probe synthesis, the coding regions of candidate genes were PCR amplified and cloned into LH95.

**Plant transformation**. The Venus-N7 reporter plasmids were introduced into *Agrobacterium tumefaciens* strain EHA105. Transformation of rice was performed as described previously[73]. Briefly, sterilized seeds were germinated on N6D medium and incubated at 29.5 °C for 3–4 weeks. The calli were subcultured on fresh N6D medium at 29.5 °C for 3 days. After infection with *A. tumefaciens* cell suspensions, the calli were washed with sterile water, transferred to N6D-S medium, and incubated at 29.5 °C. After 3-4 weeks, the resistant calli were transferred to MS-NK medium for shoot regeneration.

**Identification of rice root mutant**. The rice GATA6 CRISPR-Cas9 mutant (KO #1) was generated by Biogle Genome Editing Center[74]. The GATA6 CRISPR-Cas9 mutant (KO #2) was generated by the W.-H.L. lab (unpublished data). The homozygous mutant of *OsGATA6* was identified by PCR-based genotyping and confirmed by Sanger sequencing of PCR products. The genotyping primers were listed in Supplementary Data 8.

**Preparation of root samples for scRNA-seq**. The rice root single cells were prepared as previously described with modifications[19]. Briefly, the tip regions (1.0 cm in length from root tip) of rice radicles were harvested and digested in the RNase-free enzyme solution (4% cellulase R10, 1.5% macerozyme R10, 0.4 M mannitol, 0.1 M 4-morpholineethanesulfonic acid, 10 mM KCl, 10 mM CaCl$_2$ and 0.1% BSA) for 2 h at room temperature. The protoplasts were filtered 3–4 times with cell strainers (40 μm in diameter, Falcon, Cat No./ID: 352340), concentrated, and washed 3-4 times with 8% mannitol at room temperature. The protoplast viability was determined by trypan blue staining. The ratio of viable cells to total cells for each sample was >85%. The concentration of protoplasts was counted by a hemocytometer, and finally adjusted to 1500–2000 cells/μL.

**scRNA-seq library construction and sequencing**. Briefly, the root radicle single-cell suspensions were loaded on a Chromium Single Cell Instrument (10× Genomics, Pleasanton, CA) to generate single-cell GEMs. scRNA-seq library was generated with the Chromium Single Cell 3' Gel Bead and Library Kit v3 (10× Genomics, Cat No./ID: P/N 1000075 and 1000073) according to user guide (Chromium Single Cell 3' Reagent Kits v3, CG000183 Rev A). The qualitative analyses of DNA libraries were performed with Agilent 2100 Bioanalyzer. The libraries were sequenced by Illumina sequencer NovaSeq (Genergy Biotechnology Shanghai Co., Ltd) using two 150 bp paired-end kits. The raw scRNA-seq dataset was comprised of 28 bp Read1, 150 bp Read2 and 8 bp i7 index reads.

**ATAC-seq experiment**. ATAC-seq were performed according to published protocol[75]. Briefly, the MZ and EZ tissues were harvested and cut into small pieces in lysis buffer. The slurry was filtered and resuspended in lysis buffer. The crude nuclei were stained with 4,6-diamidino-2-phenylindole and loaded into a flow cytometer (BD biosciences, FACSAria III). The nuclei were sorted and washed once with Tris-Mg buffer. The purified nuclei were tagmented with Tn5 transposome (Vazyme Biotech, Cat No./ID: 501-02). The resultant DNAs were purified using a Qiagen MinElute PCR Purification Kit (Qiagen, Cat No./ID: 28004) and amplified using 2x NEBNext High fidelity PCR mix (New England Biolabs, Cat No./ID: M0541L). Amplified libraries were purified with AMPure beads (Beckman Coulter, Cat No./ID: A63880), and sequenced on Illumina sequencer NovaSeq by PE150 strategy. Two biological replicates were performed.

**In situ hybridization assays**. For probe synthesis, in vitro transcription was performed with T3 or T7 RNA polymerase (ThermoFisher, Cat No./ID: EP0101/EP0111) in which linearized vectors were added as templates. RNA in situ hybridization was performed as described[76]. Briefly, root tissues were harvested and fixed with formaldehyde. Paraffin-embedded samples were sectioned (7-9 μm) with a Lecia sliding microtome (RM2265). The slides were dewaxed, digested with Proteinase K (Roche, Cat No./ID: 03115828001), dehydrated with gradient ethanol, hybridized with corresponding probes and incubated with anti-digoxigenin-AP Fab fragments (Roche, Cat No./ID: 11093274910). After washing, the signals were detected with NBT/BCIP stock solution (Roche, Cat No./ID: 11681451001).,

**Microscopy**. Radicles were stained with FM4-64 (1.0 μg/mL). The fluorescence signal was observed under a Zeiss 880 upright confocal microscope. The 514 nm laser was used for excitation. For Venus, emission light wavelengths between 519 and 573 nm were detected; for FM4-64, emission was 698–759 nm.

**Pre-processing of raw scRNA-seq data**. The raw files were analyzed by Cell Ranger 3.0.1 (10x Genomics). The rice genome and GTF annotation files which exclude the organelle genomes were downloaded from the Rice Annotation Project Database (RAP-DB) website (https://rapdb.dna.affrc.go.jp/index.html). To calculate the percentage of mitochondrial genes in each cell, the mitochondrial genome was merged with the nuclear genome. The "cellranger mkref" function with "--genome, --fasta and --genes" arguments was used to build reference. The "cellranger count" with "--id, --transcriptome, --fastqs, --sample and --r2-length = 98" arguments was performed to generate single-cell gene counts. More than 90% reads in all the samples were aligned to the Nipponbare reference genome (IRGSP 1.0) by the aligner STAR (v.2.5.1b)[77]. The ratio of the number of fraction reads in cells to total number of reads for each sample was >77%. The detailed Cell Ranger reports were given in Supplementary Data 1. The gene-cell matrices (named 'filtered_gene_bc_matrices' by 10× Genomics) were served as processed raw data for further analyses.

**Doublet detection**. We identified doublets with DoubletFinder (v.2.0.2)[78]. Three input parameters, namely the number of expected real doublets (nExp), the number of artificial doublets (pN), and the neighborhood size (pK), were defined as follows: For nExp, the standard Seurat processing pipeline was performed up to the clustering stage with the low cell cluster number resolution (resolution = 0.5). The cluster labels of cells were used as "annotations" data to model the proportion of homotypic doublets. The doublet ratio was estimated by $N/100000$ ($N$, the cell numbers). The nExp value was adjusted according to the proportion of homotypic doublets and doublet ratio. pN, a ratio to define the number of generated artificial doublets based on total cell numbers, was set to 0.25. We found that an increase in pN value did not alter DoubletFinder results. To define an optimal pK value, pre-processed Seurat data were loaded into the "paramSweep_v3 (PCs = 1:15)" function, and subsequently fed into "summarizeSweep" and "find.pK" functions. A single and easily discernible maximum of pK value was selected as an optimal pK parameter.

The doublets were finally predicted with the pre-processed Seurat data using "doubletFinder_V3" function and defined values of nExp, pN, and pK as defined above. The proportion of artificial nearest neighbors (pANN) for every cell was computed. The doublet threshold of pANN was defined according to nExp to generate final doublet predictions. The resultant cells flagged as singlets were used for downstream analyses.

**Data integration, clustering, and annotation**. Downstream analyses were mainly performed with the Seurat package (v.3.1.1) as previously described[19]. Briefly, we performed quality control procedures by filtering out low-quality cells and genes, and normalizing data with "NormalizeData" function (LogNormalize method, scaling factor of 10,000). We then detected variable genes with "FindVaria-bleGenes" function (vst method, 2000 features), scaled data with "ScaleData" function, performed PCA analysis with "RunPCA" function (100 principal components), determined statistical significance of PCA scores by "JackStraw" function, constructed the SNN graph, clustered cells based on Louvain ("FindNeighbors" and "FindClusters"), and visualized data with non-linear dimensional reduction algorithms ("RunTSNE" and "RunUMAP").

The low-quality cells and genes were filtered according to the following four criteria: (1) we only considered the cells with number of expressed genes between 500 and 6000; (2) we ignored the cells with unique molecular identifiers (UMIs) above 40,000 or below 500; (3) we filtered out the genes that were expressed in fewer than three cells; (4) the percentage of mitochondrial UMIs was no more than 10%. We corrected batch effects between samples with "RunHarmony" function[27]. The resultant integrated data were clustered and visualized based on harmony dimensionality reductions.

To mitigate the effects of cell cycle heterogeneity on cell clustering, the cell cycle score of each single cell was calculated by using the "CellCycleScoring" function with the cycling orthologous genes in Arabidopsis. These cell cycle effects were then regressed out by the "ScaleData" function using "vars.to.regress".

To evaluate the effect of protoplasting genes for cell clustering, the proportion of the protoplasting genes which was identified by the Qian lab (Supplementary Data 2)[26] was calculated and plotted. The proportion of protoplasting genes in each cell cluster was below 1%.

The cluster-enriched genes were computed with "FindAllMarkers" function in Seurat using the following parameters: a Wilcoxon Rank Sum test; above 1.5-fold difference (logfc.threshold = 0.58) between the two groups of cells; test genes that a minimum fraction was at least 0.1. We assigned cell clusters by the resultant cluster-enriched genes with known functions and expression pattern (Supplementary Data 2; Supplementary Data 3). To identify cluster-specific marker genes, the following parameters were applied: the log2 fold change of genes was >0.25 and the proportion of marker genes expressed in cells among all other clusters (PCT2) was <10%.

**Inference of differentiation trajectories**. For Palantir pseudotime analysis (Fig. 2b–e; Fig. 3c, d, f; Supplementary Fig. S3b, c), we read raw data slot of the Seurat object (clusters 1, 4, and 9 for epidermis; clusters 0, 3, 6, and 11 for ground tissues) into Palantir (v.0.2.2)[31]. Data preprocessing procedures included principal component analysis, diffusion maps analysis, and MAGIC imputation. The start cell, defined based on prior and proven information and an appropriate coordinate position in $t$-SNE map, was specified before running Palantir. The terminal cell states were automatically determined. Each cell along pseudotime and branch probabilities to terminal states were assigned by Palantir. The gene expression trends along Palantir pseudotime or different lineages were modeled based on generalized additive models provided by Palantir.

The ForceAtlas2 embedding and diffusion map were performed in SCANPY (Fig. 3g; Supplementary Fig. S3d, e). Briefly, the subset of the Seurat object (clusters 0, 3, 6, and 11) was loaded into SCANPY (v.1.4.5.1)[79]. We performed downstream analyses according to user manual: data normalization (scanpy.pp. normalize_per_cell method, scaling factor 10000), log-transformation (scanpy.pp. log1p), batch correction (scanpy.pp.combat), selection of variable gene (scanpy.pp. highly_variable_genes), data scaling (scanpy.pp.regress_out), principal component analysis (scanpy.pp.pca), computation of the neighborhood graphand (sc.pp. neighbors) and visualization with ForceAtlas2 and Diffusion maps (sc.tl.diffmap, sc.pl.diffmap, sc.tl.draw_graph and sc.pl.draw_graph). The cluster label of each cell was assigned with its subclustering cluster number. For DPT pseudotime analysis, the same start cell in Palantir was used.

**RNA velocity analysis**. We estimated RNA velocity by velocyto package[33]. Expression matrix of unspliced and spliced mRNAs in each sample were generated with velocyto CLI (v.0.17.17) according to user guide (velocyto run10×). The output loom files of rice root #1 and #2 were combined using "loompy". The resultant merged loom file was read into velocyto.R (v.0.6). The unspliced and spliced counts of the cells from cluster 0, 3, 6, and 11 were extracted. We found that the reads that mapped confidently to intronic regions were <1% in rice, which is far less than 35% in human. Data normalization and variable gene detection were performed to filter out lowly variable genes. The resultant 5000 variable genes were further filtered using "filter.genes.by.cluster.expression" function (emat: min.max. cluster. average = 0.1; nmat: min.max.cluster. average = 0.01). The final 1459 genes were used to estimate RNA velocity with "gene.relative.velocity.estimates" function (kCells = 20). To plot individual cell velocities, the UMAP embeddings in Seurat were exported (Fig. 3b).

**GO analysis**. All differentially expressed genes including cluster-enriched genes, gene modules related to pseudotime analysis and the genes associated differential peaks with statistical significance were submitted to AgriGO v2.0 for GO

enrichment analysis[80]. Whole gene sets were used as the background. The top 20 GO terms with -$\log_{10}$FDR values were represented (Supplementary Data 4; Supplementary Data 6).

**ATAC-seq data analysis**. ATAC-seq data analyses were performed according to published methods with some modifications[75,81,82]. Briefly, the raw sequencing reads were first trimmed by fastp (v.0.20.0) with an adapter sequence "CTGTCTCTTATACACATCT" to obtain clean fastq files. The clean reads were aligned to the Nipponbare reference genome (IRGSP 1.0) with Bowtie2 (v.2.3.4.3)[83]. Using Samtools (v.1.9), the SAM files was converted to indexed BAM files[84]. The duplicated reads were marked and removed by sambamba (v.0.6.7) and bedtools (v.2.25.0)[85]. The biological replicates were merged by Samtools. For visualization and comparative analysis, the individual or merged datasets were converted to bigwig formats using bamCoverage provided by deepTools (v.3.1.2) with a bin size of 10 bp and normalized by Bin Per Million mapped reads (BPM). The resulting bigwig files were loaded into Integrative Genomics Viewer (IGV, v.2.4.14) for visualization. The narrow peaks were detected by the "callpeak" program in MACS2 (v.2.1.2) with a genome size of 3.6e8. The differentially accessible regions were analyzed by DiffBind (v.2.14.0) with the parameter "minOverlap = 1"[86]. For peak annotation, the rice TxDb object contained transcript annotations was created by "makeTxDbFromGFF" supplied by GenomicFeatures (v.1.38.0)[87]. The differential peaks were annotated with gene identities using ChIPseeker (v.1.22.0)[88]. The significantly accessible genes were identified with the following cutoffs: the fold change of average accessible difference ≥ 1.5 ($\log_2$FC ≥ 0.58) and FDR < 0.05.

To identify enriched transcription factor binding motifs within accessible peaks, the significantly accessible regions were scanned using "findMotifsGenome.pl" function provided by HOMER (v.4.10) with the parameter "-mset plants"[46].

For integrative analysis of ATAC-seq and scRNA-seq, the accessibilities of cell cluster-specific marker genes were extracted and assigned. For each cell cluster, the correlation between accessibility (ATAC peak $\log_2$FC) and expression level (scRNA cluster-specific $\log_2$FC) of each gene was calculated and visualized by the volcano plot (Supplementary Fig. S6). Genes with positive values for both accessibility and expression level stand for positive correlation (red). Conversely, it indicates negative correlation (blue). The number and proportion of cluster-specific genes with positive or negative correlation were then calculated (Fig. 4f).

**Identification of one-to-one orthologs**. To identify orthologous genes, the protein sequences of rice and Arabidopsis were download from the TAIR and Phytozome website. The OrthoMCL algorithm was used to cluster similar protein sequences between rice and Arabidopsis by an all-against-all strategy with BLASTP (e-value: 1-e5). The clustering results can be grouped into three subcategories: one-to-one orthologs, one-to-many groups, and many-to-many groups. Because duplicated genes usually undergo neofunctionalization, one-to-many or many-to-many groups can't be considered functionally equivalent. For this reason, only one-to-one pair orthologs were then used for cross-species scRNA-seq analysis. The gene list for one-to-one pair orthologs was given in Supplementary Data 7.

**Interspecies scRNA-seq data comparison**. Two independent strategies were adopted for cross-species analyses. First, given the fact that the rice and Arabidopsis cell atlases have been well annotated, we calculated cross-species pairwise correlation between cell types using a "gene-specificity index" equation (Fig. 5a). This method has been used to analyze the conserved cell types between turtles, lizards, and mammals[89]. Briefly, the average gene expression level of each cell type was calculated using the "AverageExpression" function in Seurat. Before calculating pairwise cell type correlations, gene expression matrices were transformed into gene specificity matrices based on "gene-specificity index" equation. The resulting gene specificity matrices were used to calculate pairwise Spearman rank order correlations. For significance analysis of correlation coefficients, gene expression values were shuffled 1000 times across cell types. The resulting Spearman correlation coefficient was calculated and termed shuffled rho value. The $p$ value was calculated as the fraction of the absolute value of shuffled values that were greater than or equal to the absolute value of rho for the non-shuffled data. The significant values ($p < 0.05$) indicated that the calculated correlations between cell types were not randomly generated (Fig. 5a).

The second strategy we used is to combine rice and Arabidopsis scRNA-seq datasets directly. To this end, we first constructed an integrated Arabidopsis root cell atlas by merging three published root scRNA-seq datasets (PRJNA517021, GSE123013 and GSE123818), from our lab[19], the Schiefelbein lab[20], and the Timmermans lab[22], respectively. The fastq files were processed by Cell Ranger 3.0.1. After quality control, a total of 23,561 cells from three scRNA-seq datasets were selected and 1,788 highly variable genes were used for downstream analysis. The Seurat workflow with similar parameters was performed as mentioned above. After clustering, 22 cell clusters (R0 to R21), which cover major Arabidopsis root cell types, were revealed (Supplementary Fig. S7a). The cell clusters were annotated by proven marker genes (Supplementary Fig. S7e; Supplementary Data 3). The batch contributions to each cell cluster are shown in Supplementary Fig. S7c, d.

The Arabidopsis and rice root scRNA-seq datasets were then integrated by Canonical Correlation Analysis (CCA) in Seurat. The batch effects across species

were removed after integration (Supplementary Fig. S8a). After clustering, 30 super cell clusters were revealed (Supplementary Fig. S8b). The comparable proportion of rice and Arabidopsis cells in a given cell type indicates cell type conservation across species (e.g., RH, P and X in Fig. 5c). To identify core regulators for conserved cell types including RH (I10), P (I21), and X (I22), we used "FindConservedMarkers" function in Seurat. The most differentially expressed genes were clustered ($k$-means = 4, Fig. 5d–f).

Note added in proof: While this work was under revision, similar research by Liu et al. was published elsewhere[90].

**Reporting summary**. Further information on research design is available in the Nature Research Reporting Summary linked to this article.

## Data availability

The scRNA-seq and ATAC-seq data were deposited in NCBI with the accession number "PRJNA706435" and "PRJNA706099". Source data are provided with this paper.

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

## Acknowledgements

We thank Yi-Qun Gao and Dai-Yin Chao (SIPPE, CAS) for discussion and Yan-Xia Mai (Core Facility Center of CEPMS/SIPPE, CAS) for technical support on the flow cytometer. This work was supported by the grants from National Natural Science Foundation of China (31788103; 31525004), Strategic Priority Research Program of the Chinese Academy of Sciences (XDB27030101), the Chinese Academy of Sciences (QYZDB-SSW-SMC002), Science and Technology Commission of Shanghai Municipality (18JC1415000), Young Elite Scientists Sponsorship Program by CAST (2016QNRC001), and National Postdoctoral Program for Innovative Talents (BX201600178).

## Author contributions

T.-Q.Z. and J.-W.W. designed the research. T.-Q.Z., Y.C., Y.L., and J.-W.W. performed research. T.-Q.Z. performed bioinformatic analysis. W.-H.L. contributed to rice *GATA6* KO #2 plants. T.-Q.Z. and J.-W.W. analyzed the data. T.-Q.Z. and J.-W.W. wrote the manuscript.

## Competing interests

The authors declare no competing interests.
