## [Peer Review File · Nature Communications]

REVIEWER COMMENTS

Reviewer #1 (Remarks to the Author):

Review of the manuscript entitled "Single-Cell Transcriptome Atlas and Chromatin Accessibility Landscape Reveal Differentiation Trajectories in the Rice Root." By Zhang et al.

In this manuscript, Zhang et al., present the transcriptome of the rice root at the single-cell level. This transcriptome complements recent analyses performed on *Arabidopsis thaliana* roots. Following cell wall digestion, Zhang et al., isolated the rice protoplasts and used them to create single-cell RNA-seq libraries. The analysis of the transcriptome allowed the annotation of many clusters and the characterization of the transcriptomic trajectories of the root epidermal cells and ground tissues. The authors applied two different strategies to annotate the rice root cells: the use of rice genes with known biological functions or expression patterns and a series of in situ hybridization using to detect the expression of 30 cluster-specific genes. To complement this analysis, Zhang et al., integrated their single-cell transcriptome with bulked ATAC-seq datasets. Finally, Zhang et al., took advantage of the release of single-cell *Arabidopsis* root transcriptomes to start analyzing similarities and differences in the activity of orthologous genes. Below are major and minor comments to the authors.

Major comments

- This reviewer enjoyed the transcriptomic analysis. However, the number of nuclei analyzed per replicate was exceptionally high leading to a low level of saturation of the transcriptome for replicate #2. Could the authors provide information about the use of a different number of cells for their analysis between replicates 1 and 2? Could the authors provide a UMAP clustering highlighting replicate 1 and replicate 2? Should another replicate be generated to provide more consistent results?
- The authors identified 4 meristematic clusters: # 5, 11, 18, and 19. Could they be associated with specific cell types based on the expression of other marker genes?
- The authors mention that they use pseudotime and embedded heatmap to characterize gene regulatory networks. I believe that the characterization of these networks would require more than a pseudotime analysis. Could the authors update this sentence to better reflect the outcome of the pseudotime analysis?
- To better analyze the role of transcription factors in controlling gene expression, the authors performed an analysis of the chromatin accessibility and the enrichment of transcription factor binding motifs specific to the meristematic and elongation zones of the root. This is one of the major weaknesses of this manuscript: performing the analysis on bulked samples from a dissected root zone might hide single-cell type specific chromatin accessibility. Single cell ATAC-seq has been applied to *Arabidopsis* nuclei (see BioRxiv manuscript from Farmer et al., 2020, and Dorrity et al., 2020 on *Arabidopsis* root nuclei). Should the authors apply similar technology on rice nuclei? Also, it seems that DAPI staining of the nuclei interferes with the activity of the transposase (<https://kb.10xgenomics.com/hc/en-us/articles/360027640311-Can-I-sort-nuclei-for-Single-Cell-ATAC-sequencing->). Did the authors perform some additional analyses to ensure that the isolation of the plant nuclei by sorting did not affect the overall accessibility of the chromatin?
- "Inter-species comparison reveals conserved and divergent root developmental pathways". The comparative analysis of *Arabidopsis* and rice is very interesting. Did the authors try to look for the conservation of expression of transcription factor genes at the single-cell level to better estimate the conservation of gene regulatory networks?

Minor comments:

- Line 35: "and markers genes" should be changed to "and marker genes"
- Line 48: Fig 1b should be annotated Fig 1a

- Lines 70-71: references must be included
- Line 78: Cell wall suberization limits the release of plant protoplasts (Schulze et al., 2019). Could the authors provide information about any limitation associated with the digestion of the rice cell wall such as limited or no access to the transcriptome of suberized rice root cells?
- Line 82: "We used standard computational pipeline" - it would be good to write here that the authors have used Cell Ranger pipeline here. This would avoid the reader looking for this basic information in other sections of the manuscript.
- Between lines 115 and 131: Figure 2 b is not mentioned in this section of the manuscript. This information could be added in line 124.
- Line 129-131 and lines 121-124 provide in some ways redundant information.
- Line 133: "We chose Os10g0452700, Os03g0155500, Os01g0248900, and Os10g0578200 as representative genes for clusters 1, 4 and 9, respectively (Fig. 2d,e; Extended Data Fig. 2b)." This sentence does not reflect the content of Fig. 2d,e, and Extended Data Fig. 2b. for instance Os01g0248900 seems to be cluster 1-specified and not cluster 4-specific, etc.. If the figure is correct than the text needs to be changed.
- Line 138: "trichoblasts" should be replaced by "atrachoblasts".
- Line 139: "atrachoblasts" should be replaced by "trichoblasts".
- Line 149: could the author include more details about the cluster 4-specific plant metabolisms referenced in the text? What are they?
- Line 190: "The above analysis revealed that OsGATA6 is expressed center of the RAM" - should be "is expressed IN THE center..."
- Line 229: Os10g0452700 does not seem to have higher accessibility in the MZ than in the EZ according to figure 4g. Extended Data Fig 6j does not confirm it.
- Lines 265-266: It would be interesting to tell more about the unique properties of the root hairs as observed by the different expression of gene clusters in Fig 5d,e,f.
- Discussion: The authors did not discuss here any of their findings, it looks more like a summary or plan. I would recommend that the authors write a real discussion and make better use of their supplementary discussion.
- Figure 1d: could the authors include more information about the cell-type specificity of the genes mentioned in this figure?
- Remarks on the Supplementary text:
Lines 20-22 on page 33 in the supplementary text: "The start cell, defined based on prior and proven information and an appropriate coordinate position in the t-SNE map, was specified before running Palantir." Could the authors give more details on how they defined the start cell?

Reviewer #2 (Remarks to the Author):

Reviewer summary:

In the manuscript entitled, "Single-cell Transcriptome Atlas and Chromatin Accessibility Landscape Reveal Differentiation Trajectories in the Rice Root" by Tian-Qi Zhang and colleagues, the authors describe a large-scale single-cell analysis of rice roots, in which they captured single-cell transcriptomes of over 25k cells from developing rice radicles. They use a combination of canonical cell type-specific gene expression patterns as well as an impressive panel of in situ hybridization experiments to assign putative cell types to each of 21 cell clusters from the resulting root atlas.

These clusters were found to be associated with an expected set of rice radicle cell types, including the exodermis and sclerenchyma cell types that have not yet been described using single-cell methodologies. The authors also perform pseudotime analyses to order epidermal and ground tissue cells, describing gene expression patterns associated with each tissue's differentiation from putative meristematic tissue. The authors go on to identify one gene associated with ground tissue differentiation as necessary for proper cell division patterns in the meristem, resulting in atypical root shape and length. The authors further perform a set of bulk-tissue ATAC-seq analysis on root meristematic and elongating cells, describing broad patterns in chromatin accessibility associated with developmental state. Lastly, the authors attempt to integrate the rice single-cell data with that from previously described Arabidopsis datasets, using a restricted set of gene orthologs between the two species. Overall, I found the manuscript to be well written, and the single-cell data described to be of adequate quality. I believe this study represents a modest advance in the plant developmental biology field with the first single-cell characterization of roots from a non-Arabidopsis species, which comes with its own set of challenges. My major criticisms have to do with the integrative analysis the authors performed with Arabidopsis data, which I found to not be entirely convincing, nor add to the overall narrative. I would strongly recommend the authors reconsider this analysis.

Specific comments:

1) While the authors provide some tabular quality control metrics for their datasets, there are some shortcomings. Firstly, how did they determine 500 UMI/cell as a lower threshold for filtering cells? Did any cells in their dataset have high numbers of mitochondrial transcripts? The methods section indicated that they aligned their reads to a version of the rice transcriptome that did not include organelle sequence. However, mitochondrial gene expression in single cells could be indicative of low RNA quality or dead/dying cells, and thus is an important quality control measure. I would recommend realigning to a complete version of the nuclear and organellar genomes, and perhaps making their filtering scheme somewhat more stringent. I also think this might help improve the reproducibility among replicate samples, which show a fair amount of heterogeneity among clusters. I would have expected two replicate samples grown under the same conditions to have a much higher degree of integration.

In addition, the authors do not mention approximately how many cells were actually loaded into each lane of the Chromium device, though they do mention rough concentrations of cells. As this has an impact on the number of doublets expected, it might be useful to include (e.g. as part of supplemental table 1).

2) The authors classify cells based on most likely cell cycle phase, yet do not mention what set of genes they used for this analysis. They also state that cell cycle is not likely driving cell clustering, yet there is a clear heterogeneity among cell clusters with respect to dividing cell proportion. Perhaps a better way to visualize the impact of cell cycle on the dimension reduction/clustering results would be to plot the actual S or G2/M scores on the UMAP axes, to identify foci that are undergoing division. I suspect that meristematic cell clusters might have a strong enrichment. I would also suggest the authors consider regressing the impact of cell cycle out of their analysis.

3) On line 86-87, the authors describe how they chose principal components based on statistical significance, but the test they used for this determination is not specified here or in the methods. Please clarify.

4) Some cell types (e.g. endodermis) are split into several clusters that are topologically separated on the main UMAP plot – some explanation as to why this might be (or an assessment of confidence for assigning each cluster a cell type) would be good to include.

5) The authors mention that they performed over 30 in situ hybridization assays for cluster-specific genes, yet only 18 are shown in the Extended Data (unless I am missing something). Were the other assays not informative?

6) For the in situ experiments, have the authors considered doing cross-sectional slices instead of longitudinal? While this would make analysis of developmental gradients difficult, it might help to distinguish expression in closely-spaced cell layers, especially for those genes that are not

expressed highly.

7) I am somewhat confused by the panel of genes that are meristem-specific. While the scRNA-seq based expression patterns and in situ clearly show these genes to be expressed in the meristematic tissues, the expression is very broad, and seems to overlap cells which should be differentiating (e.g. compare the differentiated epidermal cells in S10b, S10f, and S10i vs S10m-r). As the cells expressing the cell type-specific markers are clearly in the meristematic region, why would they not cluster with the other meristematic cells (e.g. cluster 5)? Some explanation of how to completely reconcile the in situ data with the UMAP-based expression patterns would be very useful.

8) On lines 102-104, authors should include reference to the relevant figure in the Extended Data.

9) In validating their epidermal cell lineage analysis, the authors included confocal images of genes that have informative expression patterns. However, these images are not very quantitative, making interpretation of their results difficult. For example, in figure 2e, they predict that Os10g0452700 goes down in both epidermal tissue types, yet the confocal image in 2f clearly shows cells expressing this gene in the elongation zone. The higher cell density (and thus higher PI staining) in the meristem makes comparison of the two different zones difficult. A better z-section showing specific epidermal expression patterns across developmental time, with accompanying quantitation of expression would greatly help to confirm their predictions. In addition, for Os03g0155500, Palantir results predict expression to increase in both hair and non-hair cells, yet the longitudinal confocal image only seems to show expression in non-hair cells. It might help to have several cross-sections from different zones to help to make this point.

10) I found it interesting that cluster 13, corresponding to differentiating endodermis is more proximal to the meristematic clusters than cluster 2, which the authors state has more meristematic identity. Some discussion on whether this is significant might be good to include.

11) Why did the authors not include endodermis clusters in their pseudotime analysis of the ground tissue layers? In addition, as the authors assigned cluster 8 as also belonging to the schlerenchyma layer, why did they not include this cluster in their targeted analysis?

12) In-situ analysis of OsGATA6 and OsGRF6 indicate a meristematic expression pattern, but the authors only show UMAP expression plots specific to cluster 11 and the ground tissue. Are these genes expressed more broadly across meristematic clusters (e.g. in cluster 5)?

13) I found it very promising that the authors were able to identify a gene predicted to be involved in root development that caused a clear phenotype. To better support their claim, a more quantitative analysis of the mutant phenotype should be included (potentially several mutants, or replicates of the same mutant; clearer pictures describing exactly how this gene might be involved in development).

14) My interpretation of the inter-species analysis is that it did not work as well as the authors claim. While the authors highlight a few clusters that have some mixing of cells from the different species, the majority of clusters are homogeneous to either rice or Arabidopsis. While this is to be expected for tissue types that are not present in Arabidopsis (e.g. schlerenchyma, exodermis), I would have thought most cells from Arabidopsis would be integrated in rice clusters. Furthermore, the heatmap showing correspondence between average expression in rice and Arabidopsis clusters seems to show only a weak relationship, with the exception of the meristematic clusters, with whole groups of tissues showing correlations with non-related cell types (e.g. root hairs with most ground tissues). I do not think that any real conclusions can be drawn from this analysis other than that the genes driving cell type differences are not highly represented among the 1-1 homology pairs. I would strongly suggest the authors reconsider or remove this cross-species integrative analysis, or at least discuss why the integration isn't as complete as it should be, as this analysis doesn't seem to add very much to the overall study.

15) I thought the enrichment of transporter genes among epidermal cells was very interesting. I wonder whether the authors sought to do a more comprehensive analysis of all predicted transporters in rice, and whether there is a general bias towards expression in the epidermal layers.

16) In the supplemental information, when discussing the prx gene expression patterns, the authors describe expression as overrepresented in schlerenchyma and exodermis, however there also seems to be some expression in the cortex cell layers, which should be indicated.

17) The authors identified two "vascular cylinder" cell clusters that are otherwise undefined. Do the authors have any hypotheses as to what cell types these represent? How are they different from the other more specifically defined vascular cell types?

18) For cluster 14, the authors performed an in situ hybridization experiment that indicates this to be associated with root cap junction. Why isn't this cluster assigned a cell type identity (e.g. in Fig 1)?

19) For GO term analysis, what gene set was used as the background? This should be indicated in the methods.

Aesthetic comments:

1) In figure 1, the color scheme chosen for describing each cluster's identity makes it difficult to tell cell types apart. I would suggest the authors group clusters from the same cell type using similar colors.

2) Unless there are differences being pointed out between the different dimension reduction methods (t-SNE or UMAP), I would suggest the authors stick with one or the other, as comparing e.g. Main text Figure 1 with Extended Data Figure 1 is difficult.

3) The authors do a nice job analyzing the epidermal cells identified, and clearly show clusters 1, 4, and 9 potentially correspond to epidermal progenitor, hair and non-hair cells. However, they should label these clusters as their specific identity in Figure 1, instead of making the reader flip back and forth between figures to determine identity.

4) The authors rely on connectivity in 3D space to make the argument that several ground tissue layers are connected in their dimensional reduction. However, this argument is really hard to make on paper, even plotting several angles. I would suggest the authors include either a rotating movie, or some sort of interactive plot (could be data with code to generate) to help the reader understand these relationships.

5) For Figure 4f, ordering the clusters based on whether they correspond to meristematic or elongation zones would aid interpretation.

6) For various heatmaps in the main text, a colorscale is provided with no units (goes from 0 to max). This should be changed to either describe the direction of change (i.e. with an arrow indicating "increasing expression") or with actual units.

Reviewer #3 (Remarks to the Author):

In this study the author used scRNA seq to decipher the differentiation trajectories of root meristem radicle cells in different root tissues. Root meristem protoplasts were used. Clusters were identified and assigned to different meristem root cell type using beacon genes already functionally characterized and in situ hybridization using cluster specific genes. Clusters for main root cell type were identified except for the root cap. For the epidermis root cluster, authors inferred the differentiation trajectories from the cluster corresponding to the epidermal cell meristem type to the clusters corresponding to hair or non-cell final differentiation status. This inference was experimentally validated in planta using reporter lines for a sample of 4 genes. Similarly genes expressed in the ground tissues were characterized and validated by in situ hybridization and for one of them GATA6 inactivated by CrisprCas9 mediated targeted mutagenesis. Transcriptomics cluster were found coherent with the ATAC-seq analysis of the meristematic zone and the elongation zone of the radicle meristem. Finally a comparison with arabidopsis RAM scRNAseq data base revealed for some clusters, associated with epidermis, xylem and phloem differentiation, conserved core gene list but also specie specific genes that likely reflect the specificities and divergences between monocot and dicot.

I do not have the skills to assess the relevance of the analysis and inference methods used here but I find that this article is original and provides a sum of new data which will be invaluable for the study of plant development. For this reason I am in favor of its publication. However, before this is considered, I would like the two major following points to be taken into account by the authors.

194-197: the phenotype presented for the GATA6 KO mutant (supfig4) are only qualitative and visual. I should expect here some quantification and statistical analysis of the whole plant level phenotype and also a more precise description including quantification and statistical analysis of the phenotype observed at the histological level in the RAM.

Do you obtained only one KO allele and if yes do you think it is enough to support definitive conclusion? it should be better to have at least two different alleles. Do you have also transformed no mutated segregating line for control in addition to the wild type?

I think that the presentation and the analysis of the GATA6 phenotype should be strongly improved to support the conclusion "these findings substantiate the precision of scRNA-seq in identification of novel rice root mutants at the cell-specific level." that constitutes a key point of this article.

In the conclusion it is claimed that this work may help identify key genes involved in plant adaptive development. As mentioned in the introduction the major part of the rice root system is constituted by adventitious roots and the adaptive development of the rice root system is based on the modulation of adventitious and/or lateral root initiation or development. For this reason I think that in addition to compare the newly scRNA seq data with similar dataset obtained in Arabidopsis, it should be interesting to know as far your obtained data overlap with exhaustive transcriptomic data obtained during initiation (Lavarenne et al., 2019, Plant Journal 100, 954-968) or in different zones and tissues (Takehisa et al., Plant J. 2012;69: 126-140) of rice adventitious roots. In particular in this last study there is a root cap specific gene list that could help to identify a root cap cluster in your data sets. In addition I think this comparison between embryonic and post embryonic rice root transcriptomes will be interesting for the scientific community.

Another important point 18, 76 and after : The rice primary root must be named radicle not radical

two minor points:

29: "... is therefore crucial for future crop design."

I don't think this is the most important perspective of this work to be highlighted. The most important newly perspective for me is relative to a better understanding of RAM functioning in particular in an evolutionary point of view between dicot and monocot

190 "... is expressed [in the] center of the RAM...."

Reviewer #4 (Remarks to the Author):

Unlike Arabidopsis, which the molecular basis of root development and setting up root cell identity is well studied; that in rice is still obscure. To understand the developmental trajectories and transcriptional networks in rice root, the authors conducted scRNA sequencing and chromatin accessibility survey of rice radicles. Further analysis showed that the rice root tip is composed of highly heterogeneous cells. Different clusters can be assigned to 10 major root cell types according to the scRNA-seq data. They further showed that this analysis could help to identify new players for root development in rice. Overall, the authors presented a novel rice root single cell transcriptome and revealed the conserved and divergent root developmental pathways between rice and Arabidopsis, which will be of great value for the field. However, there are a few concerns need to be clarified.

1. First of all, the authors need to give a brief description of the quality of scRNA-seq data, for example, the median number of genes and transcripts detected per cell. Genes induced by protoplasting should be removed prior to analysis.

2. Could the authors explain the absence of the well-known transcription factors for root development, like Scarecrow and PLT etc. Furthermore, it's quite strange that the authors could detect the expression of OsSHR1 and OsRHL1 in sclerenchyma. Both have been showed to be absent in these cells (Cui et al., 2007, Ding et al., 2009). This information needs to be carefully verified.

3. As hormones play important role in the root development, it might be better for the authors to analyze the pattern of hormone synthesis and responses in different cell clusters.

4. Because the authors did not detect QC cells in their scRNA-seq data, this information should be discussed. Furthermore, did they remove the dataset of QC cell cluster in the inter-species comparison analysis?

5. In line 155-160, the description of the development of ground tissue is a bit miss leading. Normally, the ground tissue in rice consists of exodermis, sclerenchyma, cortex and endodermis (Rebouillat et al., 2009). A detailed description of formative cell division pattern in rice ground tissue stem cells can be found in Ni et al., 2014 (Plant Biology). The authors need to revise the description.

Reviewer #1 (Remarks to the Author):

In this manuscript, Zhang et al., present the transcriptome of the rice root at the single-cell level. This transcriptome complements recent analyses performed on *Arabidopsis thaliana* roots. Following cell wall digestion, Zhang et al., isolated the rice protoplasts and used them to create single-cell RNA-seq libraries. The analysis of the transcriptome allowed the annotation of many clusters and the characterization of the transcriptomic trajectories of the root epidermal cells and ground tissues. The authors applied two different strategies to annotate the rice root cells: the use of rice genes with known biological functions or expression patterns and a series of in situ hybridization using to detect the expression of 30 cluster-specific genes.

To complement this analysis, Zhang et al., integrated their single-cell transcriptome with bulked ATAC-seq datasets. Finally, Zhang et al., took advantage of the release of single-cell *Arabidopsis* root transcriptomes to start analyzing similarities and differences in the activity of orthologous genes. Below are major and minor comments to the authors.

Thanks for your supportive comments.

Major comments

1. This reviewer enjoyed the transcriptomic analysis. However, the number of nuclei analyzed per replicate was exceptionally high leading to a low level of saturation of the transcriptome for replicate #2. Could the authors provide information about the use of a different number of cells for their analysis between replicates 1 and 2? Could the authors provide a UMAP clustering highlighting replicate 1 and replicate 2? Should another replicate be generated to provide more consistent results?

Re:

1) For scRNA-seq, it is general accepted that the more cells, the higher chance to identify rare cell types. Although the cells vary between two replicates, the resulting number of cell clusters is the same (Supplementary Fig. 1e-g). Thus, this result suggests that 1) the number of cells in the replicate #1 is already saturated; 2) the number of sequenced cells does not affect our conclusion; 3) the inferred number of cell clusters is reliable.

2) The sequencing depth does not affect cell clustering. As you can see in Supplementary Table 1, we actually got more genes per cell in the replicate #2. Please also note that we used the 10x

Genomics Kit v3 for replicate #2. Generally, the 10x Genomics Kit v3 was much effective than the v2 version in capturing cells and genes.

3) In response to your request, we provided a UMAP colored by samples in Supplementary Fig. 1f.

Figure R1-1. UMAP showing the reproducibility between two biological replicates.

See also Supplementary Fig. 1f.

2. The authors identified 4 meristematic clusters: # 5, 11, 18, and 19. Could they be associated with specific cell types based on the expression of other marker genes?

Re: The meristematic cells usually have low cell heterozygosity. As such, these four cell clusters (i.e. clusters 5, 11, 18, and 19) do not differ greatly in their transcriptomes. However, we have shown that cluster 11 serves as meristematic cells for ground tissues (Fig. 3).

3. The authors mention that they use pseudotime and embedded heatmap to characterize gene regulatory networks. I believe that the characterization of these networks would require more than a pseudotime analysis. Could the authors update this sentence to better reflect the outcome of the pseudotime analysis?

Re: We agree. We revised this sentence “To obtain a better understanding of the gene regulatory **Basis** underlying ground tissue differentiation, we re-clustered the cells from clusters 0, 3, 6 and 11, and performed pseudotime and embedded heatmap analyses.”.

4. To better analyze the role of transcription factors in controlling gene expression, the authors performed an analysis of the chromatin accessibility and the enrichment of transcription factor binding motifs specific to the meristematic and elongation zones of the root.

This is one of the major weaknesses of this manuscript: performing the analysis on bulked samples from a dissected root zone might hide single-cell type specific chromatin accessibility. Single cell ATAC-seq has been applied to Arabidopsis nuclei (see BioRxiv manuscript from Farmer et al., 2020, and Dorrity et al., 2020 on Arabidopsis root nuclei). Should the authors apply similar technology on rice nuclei? Also, it seems that DAPI staining of the nuclei interferes with the activity of the transposase (<https://kb.10xgenomics.com/hc/en-us/articles/360027640311-Can-I-sort-nuclei-for-Single-Cell-ATAC-sequencing->). Did the authors perform some additional analyses to ensure that the isolation of the plant nuclei by sorting did not affect the overall accessibility of the chromatin?

Re:

1) scATAC-seq. Thanks for your suggestion. Since the application of scATAC-seq assay in rice root has not been reported, we think that the requested experiment is a bit beyond the scope of our manuscript. It will take at least one or two years for us to establish a suitable experimental pipeline for this assay. Indeed, we have shown that the combination of bulk ATAC-seq and scRNA-seq could still identify cell type-specific TF binding sites in the rice root tip. For example, the genes (Os10g0452700 and OsEXPA11) in cluster 9 (meristematic cells corresponding to epidermal cells) exhibited higher accessibility in the MZ than in the EZ (Fig. 4g; Supplementary Fig. 6j). In contrast, the genes (Os08g0112300, Os10g0490900, Os07g0176600 and OsPKS16) enriched in clusters 1 and 4 (corresponding to differentiated epidermal cells) were largely inaccessible in the MZ but became accessible in the EZ (Fig. 4g; Supplementary Fig. 6b,e). Future work using Chromium Next GEM Single Cell Multiome ATAC + Gene Expression system (<https://www.10xgenomics.com/product-list/#multiome>) will shed light on how TFs contribute to cell type specification in rice roots.

2) The effect of DAPI staining on ATAC-seq. Lu et al. has demonstrated a nice correlation among FANS (fluorescence-activated nuclei sorting)-ATAC-seq, DNase-seq (without DAPI staining) and ATAC-seq (without DAPI staining) (Nucleic Acids Research, 2017 / <https://doi.org/10.1093/nar/gkw1179>). Similarly, our recent work has shown the reliability of nuclei sorting-based ATAC-seq assay in Arabidopsis (Wang et al., Dev Cell, 2020 / <https://doi.org/10.1016/j.devcel.2020.07.003>). Consistent with these findings, our ATAC-seq data are reliable: we have obtained 19,872 differentially accessible peaks (FDR < 0.05; log₂(fold

change) > 0.58 or < -0.58). Notably, our data could be further validated by previous knowledge. For example, the meristematic genes, *OsPLT1* and *OsPLT5*, harbors higher chromatin accessibility in the MZ than in EZ (see below, see also Fig. 4b). The effect of DAPI staining of the nuclei on the activity of the transposase DAPI could be only observed in the scATAC-seq experiment.

Figure R1-2. The chromatin accessibility of *OsPLT1* and *OsPLT5* loci in the MZ and EZ.

5. “Inter-species comparison reveals conserved and divergent root developmental pathways”. The comparative analysis of Arabidopsis and rice is very interesting. Did the authors try to look for the conservation of expression of transcription factor genes at the single-cell level to better estimate the conservation of gene regulatory networks?

Re: Thanks for your supportive comments. We now provide a list of transcription factor genes which are expressed in RH, X and P cell clusters (Supplementary Table 9). These genes may lay the groundwork for future characterization of conserved root development pathways.

Minor comments:

6. Line 35: “and markers genes” should be changed to “and marker genes”

Re: Fixed.

7. Line 48: Fig 1b should be annotated Fig 1a.

Re: Here we want to introduce the anatomy of rice RAM that is comprised of a stem cell niche (SCN) and undifferentiated small dividing cells.

8. Lines 70-71: references must be included.

Re: Fixed. We cited all the papers (ref. 11 to 16). We wrote “Several scRNA-seq studies have revealed that Arabidopsis root tip cells have highly heterogeneous transcriptomes¹¹⁻¹⁶.”

9. Line 78: Cell wall suberization limits the release of plant protoplasts (Schulze et al., 2019). Could the authors provide information about any limitation associated with the digestion of the rice cell wall such as limited or no access to the transcriptome of suberized rice root cells?

Re: This is a very good point. Actually, we could not guarantee that all the suberized cells were captured. We only ensured the completeness of cell wall digestion by examination of the leftover under a microscope. In agreement with your assumption, we could not faithfully identified a cell cluster correspond to the root cap. Future working using single-nucleus RNA-seq (snRNA-seq) could inform us whether some suberized cells are indeed missed in our dataset. Nevertheless, we do identify the cell clusters corresponding to the sclerenchyma layer and exodermis in our atlas.

10. Line 82: “We used standard computational pipeline” - it would be good to write here that the authors have used Cell Ranger pipeline here. This would avoid the reader looking for this basic information in other sections of the manuscript.

Re: Fixed.

11. Between lines 115 and 131: Figure 2 b is not mentioned in this section of the manuscript. This information could be added in line 124.

Re: Thanks. It is fixed.

12. Line 129-131 and lines 121-124 provide in some ways redundant information.

Re: Line 121-124 described the topological feature on the UMAP, whereas line 129-131 summarized the differentiation trajectory of cluster 9.

13. Line 133: “We chose Os10g0452700, Os03g0155500, Os01g0248900, and Os10g0578200 as representative genes for clusters 1, 4 and 9, respectively (Fig. 2d,e; Extended Data Fig. 2b).” This sentence does not reflect the content of Fig. 2d,e, and Extended Data Fig. 2b. for instance

Os01g0248900 seems to be cluster 1-specified and not cluster 4-specific, etc.. If the figure is correct than the text needs to be changed.

Re: It is a typo. It has been fixed. We wrote “We chose Os10g0452700, Os03g0155500, Os01g0248900, and Os10g0578200 as representative genes for clusters 9, 1 and 4, respectively (Fig. 2d, e; Supplementary Fig. 2b).”.

14. Line 138: “trichoblasts” should be replaced by “atrichoblasts”.

Re: Thanks. It is a typo. It has been fixed.

15. Line 139: “atrichoblasts” should be replaced by “trichoblasts”.

Re: Thanks. It is a typo. It has been fixed.

16. Line 149: could the author include more details about the cluster 4-specific plant metabolisms referenced in the text? What are they?

Re: Fixed. It is related to “fatty acid metabolic process”. Please see Supplementary Table 4 for details.

17. Line 190: “The above analysis revealed that OsGATA6 is expressed center of the RAM” - should be “is expressed IN THE center...”

Re: Fixed.

18. Line 229: Os10g0452700 does not seem to have higher accessibility in the MZ than in the EZ according to Fig. 4g. Extended Data Fig. 6j does not confirm it.

Re: The accessibility of *Os10g0452700* in the MZ is 1.6-fold higher than in the EZ. The p-value is 0.00528.

19. Lines 265-266: It would be interesting to tell more about the unique properties of the root hairs as observed by the different expression of gene clusters in Fig 5d,e,f.

Re: We now provide a gene list for the cell clusters in Fig 5d,e,f (Supplementary Table 9).

20. Discussion: The authors did not discuss here any of their findings, it looks more like a summary or plan. I would recommend that the authors write a real discussion and make better use of their supplementary discussion.

Re: Thanks. We have combined the supplementary discussion.

21. Figure 1d: could the authors include more information about the cell-type specificity of the genes mentioned in this figure?

Re: It is actually given in the Supplementary Table 3. We now clarify it in the legend “The full names of selected genes are given in Supplementary Table 3.”.

23. Lines 20-22 on page 33 in the supplementary text: “The start cell, defined based on prior and proven information and an appropriate coordinate position in the *t*-SNE map, was specified before running Palantir.” Could the authors give more details on how they defined the start cell?

Re: The start cells are defined by the position clusters 1, 4 and 9 on the *t*-SNE map: Since cluster 1 bifurcated into two lineages (clusters 4 and 9), the cells in cluster 1 should be assigned as start cells. For Fig. 3c,d,f, the start cell is defined by the position clusters 0, 3, 6 and 11 on the *t*-SNE map: cluster 11 is branched into three lineages (clusters 0, 3 and 6). In addition, cluster 11 was assigned as meristematic cell cluster.

Reviewer #2 (Remarks to the Author):

In the manuscript entitled, “Single-cell Transcriptome Atlas and Chromatin Accessibility Landscape Reveal Differentiation Trajectories in the Rice Root” by Tian-Qi Zhang and colleagues, the authors describe a large-scale single-cell analysis of rice roots, in which they captured single-cell transcriptomes of over 25k cells from developing rice radicles. They use a combination of canonical cell type-specific gene expression patterns as well as an impressive panel of in situ hybridization experiments to assign putative cell types to each of 21 cell clusters from the resulting root atlas. These clusters were found to be associated with an expected set of rice radicle cell types, including the exodermis and sclerenchyma cell types that have not yet been described using single-cell methodologies. The authors also perform pseudotime analyses to order epidermal and ground tissue cells, describing gene expression patterns associated with each tissue’s differentiation from putative meristematic tissue. The authors go on to identify one gene associated with ground tissue differentiation as necessary for proper cell division patterns in the meristem, resulting in atypical root shape and length. The authors further perform a set of bulk-tissue ATAC-seq analysis on root meristematic and elongating cells, describing broad patterns in chromatin accessibility associated with developmental state. Lastly, the authors attempt to integrate the rice single-cell data with that from previously described Arabidopsis datasets, using a restricted set of gene orthologs between the two species. Overall, I found the manuscript to be well written, and the single-cell data described to be of adequate quality. I believe this study represents a modest advance in the plant developmental biology field with the first single-cell characterization of roots from a non-Arabidopsis species, which comes with its own set of challenges. My major criticisms have to do with the integrative analysis the authors performed with Arabidopsis data, which I found to not be entirely convincing, nor add to the overall narrative. I would strongly recommend the authors reconsider this analysis.

Re: Thanks for your supportive comments and constructive suggestions. We have revised the manuscript according to these suggestions. For the inter-species scRNA-seq analysis, we can remove this part of results if other three reviewers and editor recommend (see below).

1. While the authors provide some tabular quality control metrics for their datasets, there are some shortcomings. Firstly, how did they determine 500 UMI/cell as a lower threshold for filtering cells?

Did any cells in their dataset have high numbers of mitochondrial transcripts? The methods section indicated that they aligned their reads to a version of the rice transcriptome that did not include organelle sequence. However, mitochondrial gene expression in single cells could be indicative of low RNA quality or dead/dying cells, and thus is an important quality control measure. I would recommend realigning to a complete version of the nuclear and organellar genomes, and perhaps making their filtering scheme somewhat more stringent. I also think this might help improve the reproducibility among replicate samples, which show a fair amount of heterogeneity among clusters. I would have expected two replicate samples grown under the same conditions to have a much higher degree of integration.

In addition, the authors do not mention approximately how many cells were actually loaded into each lane of the Chromium device, though they do mention rough concentrations of cells. As this has an impact on the number of doublets expected, it might be useful to include (e.g. as part of supplemental table 1).

Re: We agree.

1) For setting 500 UMI/cell as a lower threshold for filtering cells. Actually, this setting is higher than the standard threshold recommended by the Seurat (200 UMI/cell). The reason why we chose a high stringent threshold is to filter out the cells of low quality, thereby mitigating the effect of these cells in cell clustering.

2) For the OsRoot #1 sample, we loaded ~ 20,000 cells, For the OsRoot #2 sample, we loaded ~ 32,000 cells. We have included these information in the Supplementary Table 1.

3) As suggested by the reviewer, we downloaded rice mitochondrial genome and integrated this sequence into the nuclear genome. We then re-performed Cell Ranger analyses using previous settings. To evaluate the effect of mitochondrial genes on cell clustering, we calculate the percentage of mitochondrial genes in each single cell (see below, Figure R2-1; Supplementary Fig. 1a, Supplementary Fig. 1c). As you can see, 27463 out of 27469 cells have the percentage of mitochondrial gene lower than 5%. The percentage in the rest six cells is below 10%. Therefore, we concluded that the filtering settings we previously used has already mitigate mitochondrial genes on the atlas. We stated this analysis in the revision.

We wrote “The protoplasting and mitochondrial genes had little effects on the clustering (see Method; Supplementary Fig. 1a,c)”.

Figure R2-1. The percentage of mitochondrial genes in each single cell.

Y-axis, the percentage of mitochondrial genes in each single cell. Dot, single cell.

See also Supplementary Fig. 1a.

2. The authors classify cells based on most likely cell cycle phase, yet do not mention what set of genes they used for this analysis. They also state that cell cycle is not likely driving cell clustering, yet there is a clear heterogeneity among cell clusters with respect to dividing cell proportion. Perhaps a better way to visualize the impact of cell cycle on the dimension reduction/clustering results would be to plot the actual S or G2/M scores on the UMAP axes, to identify foci that are undergoing division. I suspect that meristematic cell clusters might have a strong enrichment. I would also suggest the authors consider regressing the impact of cell cycle out of their analysis.

Re:

1) We did not classify cells by the cell cycle phase. Actually, we have regressed out cell cycle effect in our analysis. To do this, we calculated cell cycle phase score for each single cell by the homologues cell cycle genes from Arabidopsis (The gene list is given in Supplementary Table 7). We then regressed out cell cycle effects by the "ScaleData" function with "vars.to.regress". We now describe the procedure in the Method section.

We wrote "To mitigate the effects of cell cycle heterogeneity on cell clustering, the cell cycle score of each single cell were calculated by using the "CellCycleScoring" function with the cycling orthologous genes in Arabidopsis. These cell cycle effects were then regressed out by the "ScaleData" function using "vars.to.regress".".

2) Indeed, nearly all the dividing cells mostly reside in the meristematic cell clusters (Figure R2-2).

Figure R2-2. UMAP plot showing the expression pattern of cell cycle genes.

3. On line 86-87, the authors describe how they chose principal components based on statistical significance, but the test they used for this determination is not specified here or in the methods. Please clarify.

Re: We used "JackStraw" and "ScoreJackStraw" functions to calculate statistical significance of PCA scores. The resultant 100 principal components (PCs, $p < 0.05$) were then used for downstream analyses. We now describe the procedure by which we chose PCs in the Method section.

We wrote "We then detected variable genes with "FindVariableGenes" function (vst method, 2000 features), scaled data with "ScaleData" function, performed PCA analysis with "RunPCA" function (100 principal components), determined statistical significance of PCA scores by "JackStraw" function, constructed the SNN graph, clustered cells based on Louvain ("FindNeighbors" and "FindClusters"), and visualized data with non-linear dimensional reduction algorithms ("RunTSNE" and "RunUMAP").".

4. Some cell types (e.g. endodermis) are split into several clusters that are topologically separated on the main UMAP plot – some explanation as to why this might be (or an assessment of confidence for assigning each cluster a cell type) would be good to include.

Re: Fully agree. We now describe these observations in the Supplementary text. The reason why they are separated on the UMAP is because endodermal cells involved in CS formation (cluster 13) feature a distinct transcriptome.

We wrote: “Consistently, clusters 2 and 13 were topologically separated on the UMAP. The association of cluster 13 with cluster 5 suggests that this cluster may retain some meristematic activity.”

5. The authors mention that they performed over 30 *in situ* hybridization assays for cluster-specific genes, yet only 18 are shown in the Extended Data (unless I am missing something). Were the other assays not informative?

Re: The rest of the genes did not give reliable *in situ* signals.

We wrote “We have examined over 30 cluster-specific genes. However, some of them did not give reliable signals (data not shown).” (Supplementary Fig. 10).

6. For the *in situ* experiments, have the authors considered doing cross-sectional slices instead of longitudinal? While this would make analysis of developmental gradients difficult, it might help to distinguish expression in closely-spaced cell layers, especially for those genes that are not expressed highly.

Re: Fixed. We now include more cross-sectioned *in situ* for these genes (Figure R2-3; Supplementary Fig. 10s-x). Based on these results, we can conclude that *Os01g0914100* and *Os03g0135700* are expressed in the cortex, *Os10g0155100* was expressed in the endodermis; *Os07g0634400* was expressed in the pericycle; and both *Os08g0489300* and *Os07g0638500* were expressed in the xylem.

Figure R2-3. RNA *in situ* hybridization assays on cluster-specific genes.

See also Supplementary Fig. 10s-x.

7. I am somewhat confused by the panel of genes that are meristem-specific. While the scRNA-seq based expression patterns and *in situ* clearly show these genes to be expressed in the meristematic tissues, the expression is very broad, and seems to overlap cells which should be

differentiating (e.g. compare the differentiated epidermal cells in S10b, S10f, and S10i vs S10m-r). As the cells expressing the cell type-specific markers are clearly in the meristematic region, why would they not cluster with the other meristematic cells (e.g. cluster 5)? Some explanation of how to completely reconcile the *in situ* data with the UMAP-based expression patterns would be very useful.

Re: I am sorry for this confusion. Each cell type in the root tip is composed of meristematic cells, dividing cells (i.e. transit-amplifying cells) and differentiated cells. The unsupervised cell clustering method sorts all the single cells into distinct cell clusters by their similarities in transcriptome. As such, all the dividing cells in the root tip tend to group together (i.e. clusters 5, 11, 18, 19), regardless which cell types they belong to. Nevertheless, these meristematic cell clusters also differ in some aspects. For instance, cluster 11 serves as the meristematic cells for ground tissues (Fig. 3). We now clarify this point in the Supplementary text.

8. On lines 102-104, authors should include reference to the relevant figure in the Extended Data.

Re: Fixed. We wrote “Second, we performed RNA *in situ* hybridization assays for over 30 cluster-specific genes (see below, Supplementary Fig. 10).”.

9. In validating their epidermal cell lineage analysis, the authors included confocal images of genes that have informative expression patterns. However, these images are not very quantitative, making interpretation of their results difficult. For example, in figure 2e, they predict that Os10g0452700 goes down in both epidermal tissue types, yet the confocal image in 2f clearly shows cells expressing this gene in the elongation zone. The higher cell density (and thus higher PI staining) in the meristem makes comparison of the two different zones difficult. A better z-section showing specific epidermal expression patterns across developmental time, with accompanying quantitation of expression would greatly help to confirm their predictions. In addition, for Os03g0155500, Palantir results predict expression to increase in both hair and non-hair cells, yet the longitudinal confocal image only seems to show expression in non-hair cells. It might help to have several cross-sections from different zones to help to make this point.

Re: Fully agree. We now provide more pics for the *Os10g0452700* reporter in Supplementary Fig. 5c. These pics clearly show that *Os10g0452700* was highly expressed in the epidermis in the meristematic zone and progressively declined along the developmental process (Figure R2-4).

For *Os03g0155500* reporter, we did more transverse sections (Fig. 2g). It is clear that *Os03g0155500* reporter was expressed in all epidermal cells, which is consistent with our scRNA-seq analysis results.

Figure R2-4. The expression pattern of *Os10g0452700* reporter along developmental process of rice root.

See also Supplementary Fig. 2c.

10. I found it interesting that cluster 13, corresponding to differentiating endodermis is more proximal to the meristematic clusters than cluster 2, which the authors state has more meristematic identity. Some discussion on whether this is significant might be good to include.

Re: Fully agree. This expression pattern suggests that 1) there are two types of endodermis in rice root; 2) cluster 13 cells may retain some meristematic property. We now discuss this point in the Supplementary text.

We wrote “Consistently, clusters 2 and 13 were topologically separated on the UMAP. The association of cluster 13 with cluster 5 suggests that this cluster may retain some meristematic activity.”.

11. Why did the authors not include endodermis clusters in their pseudotime analysis of the ground tissue layers? In addition, as the authors assigned cluster 8 as also belonging to the sclerenchyma layer, why did they not include this cluster in their targeted analysis?

Re: 1) Cluster 13 (En) was not topologically associated with cluster 11 (Ms) (Fig. 1c; Supplementary Movie 1). This pattern indicates that the cells belonging to cluster 13 is not derived from cluster 11. As such, we did not include this cluster in our pseudotime analysis. 2) For pseudotime analysis, we aimed to infer the early differentiation trajectory of these ground tissue layers. Since cluster 8 (sclerenchyma) stands for the mature sclerenchyma layer cells, we did not include this cluster in the analysis.

12. In-situ analysis of *OsGATA6* and *OsGRF6* indicate a meristematic expression pattern, but the authors only show UMAP expression plots specific to cluster 11 and the ground tissue. Are these genes expressed more broadly across meristematic clusters (e.g. in cluster 5)?

Re: We now include a UMAP plot showing the expression pattern of *OsGATA6* and *OsGRF6* in the atlas (Supplementary Fig. 4b).

13. I found it very promising that the authors were able to identify a gene predicted to be involved in root development that caused a clear phenotype. To better support their claim, a more quantitative analysis of the mutant phenotype should be included (potentially several mutants, or replicates of the same mutant; clearer pictures describing exactly how this gene might be involved in development).

Re: We agree with this argument. To confirm the phenotype of the *Osgata6* mutants, we studied the root phenotypes of another knock-out line (OsGATA6 KO #2) which was generously provided by Dr. Wen-Hui Lin (SJTU). The quantitative analysis of root lengths and RAM size clearly revealed an decreased meristematic activity in the *Osgata6* mutants (Figure R2-5; Supplementary Fig. 4).

Figure R2-5. The root phenotype of the *Osgata6* mutants.

See also Supplementary Fig. 4.

14. My interpretation of the inter-species analysis is that it did not work as well as the authors claim. While the authors highlight a few clusters that have some mixing of cells from the different species, the majority of clusters are homogeneous to either rice or Arabidopsis. While this is to be expected for tissue types that are not present in Arabidopsis (e.g. sclerenchyma, exodermis), I

would have thought most cells from Arabidopsis would be integrated in rice clusters. Furthermore, the heatmap showing correspondence between average expression in rice and Arabidopsis clusters seems to show only a weak relationship, with the exception of the meristematic clusters, with whole groups of tissues showing correlations with non-related cell types (e.g. root hairs with most ground tissues). I do not think that any real conclusions can be drawn from this analysis other than that the genes driving cell type differences are not highly represented among the 1-1 homology pairs. I would strongly suggest the authors reconsider or remove this cross-species integrative analysis, or at least discuss why the integration isn't as complete as it should be, as this analysis doesn't seem to add very much to the overall study.

Re: Thanks for your discussion.

1) Monocots branch out from dicots about 140-150 million years ago. The rice roots differ from those in Arabidopsis in their architecture and cell types. Thus, we would expect that there will be big difference in cell clusters between monocot root and dicot roots. In agreement with this, we found that a relatively weak relationship of most of the cell types, with the exception of the meristematic clusters. Nevertheless, we found that rice and Arabidopsis do share high similarity in root hair, phloem and xylem cell clusters (Root hair cluster: 48.3% from OsRoot cell and 51.7% from AtRoot cell; Phloem cluster: 46.3% from OsRoot cell and 53.7% from AtRoot cell; Xylem cluster: 42.8% from OsRoot cell and 57.2% from AtRoot cell; see Fig. 5c). As such, we were able to infer common and distinct genes for these three cell types (Fig. 5d-f). We agree with the reviewer's argument that other cell clusters do show weak relationship. Therefore, we revised the main text. We wrote "Pairwise comparisons of root cell clusters of Arabidopsis and rice revealed a **relatively** high degree of similarities in the clusters corresponding to meristematic, epidermis (nonhair and root hair cells), phloem and xylem cells (Fig. 5a).".

2) The method we used has been applied for the integrative scRNA-seq analysis in turtles, lizards and mammals (Tosches et al., Science, 2018 / DOI: 10.1126/science.aar4237). As shown in their Fig 5a, the correlation coefficients of cell types among turtles, lizards, and mammals are between -0.2 ~ 0.2 and -0.49 ~ 0.55, which is very similar to the correlation coefficient we got for the comparison between rice and Arabidopsis (-0.27 ~ 0.53).

3) We can remove this part of result if you and editor suggest.

15. I thought the enrichment of transporter genes among epidermal cells was very interesting. I wonder whether the authors sought to do a more comprehensive analysis of all predicted transporters in rice, and whether there is a general bias towards expression in the epidermal layers.

Re: Actually, we have surveyed about 373 rice transporter genes on the UMAP. Most of them do not harbor cell type specific expression pattern (data not shown). We have shown all the epidermal cell specific transporter genes in Supplementary Fig. 11.

16. In the supplemental information, when discussing the *prx* gene expression patterns, the authors describe expression as overrepresented in sclerenchyma and exodermis, however there also seems to be some expression in the cortex cell layers, which should be indicated.

Re: We agree. We now describe that some *prx* genes are also expressed in the cortex cell layer in the Figure legend (Supplementary Fig. 12).

We wrote “For example, *PRX54* and *PRX5* were highly expressed in cluster 13 (endodermis), whereas *PRX86*, *PRX111* and *PRX112* were predominantly expressed in cluster 2 (endodermis). In contrast, *PRX27*, *PRX32* and *PRX74* and *PRX86* were detected in cluster 0 (cortex). These expression patterns suggest that *PRXs* may exert different roles in distinct cell types.”.

17. The authors identified two “vascular cylinder” cell clusters that are otherwise undefined. Do the authors have any hypotheses as to what cell types these represent? How are they different from the other more specifically defined vascular cell types?

Re: Actually, we could not annotate these clusters. This is largely due to functional genomics study in rice is far behind those in Arabidopsis. The GO term analysis results were provided in Supplementary Table 4.

18. For cluster 14, the authors performed an *in situ* hybridization experiment that indicates this to be associated with root cap junction. Why isn't this cluster assigned a cell type identity (e.g. in Fig 1)?

Re: Thanks for your suggestion. We now annotate cluster 14 as root cap junction (see Fig. 1a and Supplementary text).

19. For GO term analysis, what gene set was used as the background? This should be indicated in the methods.

Re: We used all cluster-enriched genes as the background. We have included this in the Method section.

Aesthetic comments:

20. In Figure 1, the color scheme chosen for describing each cluster's identity makes it difficult to tell cell types apart. I would suggest the authors group clusters from the same cell type using similar colors.

Re: Thanks for your suggestion. We highlighted the same cell type using dotted lines (Fig. 1a).

21. Unless there are differences being pointed out between the different dimension reduction methods (*t*-SNE or UMAP), I would suggest the authors stick with one or the other, as comparing e.g. Main text Figure 1 with Extended Data Figure 1 is difficult.

Re: I am sorry for this confusion. UMAP is a nonlinear dimensionality reduction method. Compared to *t*-SNE, UMAP offers faster runtime and consistency, meaningful organization of cell clusters and preservation of continuums (Becht et al., 2019). In contrast, the main advantage of *t*-SNE is the ability to preserve local structure. Therefore, we used two dimensionality reduction techniques for data visualization in the manuscript. We state this reason in the main text.

Reference: Becht et al., (2019). Dimensionality reduction for visualizing single-cell data using UMAP. *Nature Biotechnology*. 37, 38–44.

22. The authors do a nice job analyzing the epidermal cells identified, and clearly show clusters 1, 4, and 9 potentially correspond to epidermal progenitor, hair and non-hair cells. However, they should label these clusters as their specific identity in Figure 1, instead of making the reader flip back and forth between figures to determine identity.

Re: Thanks for your suggestion. We have labeled these cell clusters (Fig. 1a).

23. The authors rely on connectivity in 3D space to make the argument that several ground tissue

layers are connected in their dimensional reduction. However, this argument is really hard to make on paper, even plotting several angles. I would suggest the authors include either a rotating movie, or some sort of interactive plot (could be data with code to generate) to help the reader understand these relationships.

Re: Thanks for your suggestion. We now provide a rotating movie in the Supplementary Movie 1.

24. For Figure 4f, ordering the clusters based on whether they correspond to meristematic or elongation zones would aid interpretation.

Re: Thanks for your suggestion. It is fixed (Fig. 4f).

25. For various heatmaps in the main text, a color scale is provided with no units (goes from 0 to max). This should be changed to either describe the direction of change (i.e. with an arrow indicating “increasing expression”) or with actual units.

Re: Thanks for your suggestion. We provided the directionality of the color scale by labeling “low” and “high”.

Reviewer #3 (Remarks to the Author):

In this study the author used scRNA seq to decipher the differentiation trajectories of root meristem radicle cells in different root tissues. Root meristem protoplasts were used. Clusters were identified and assigned to different meristem root cell type using beacon genes already functionally characterized and in situ hybridization using cluster specific genes. Clusters for main root cell type were identified except for the root cap. For the epidermis root cluster, authors inferred the differentiation trajectories from the cluster corresponding to the epidermal cell meristem type to the clusters corresponding to hair or non-cell final differentiation status. This inference was experimentally validated in planta using reporter lines for a sample of 4 genes. Similarly genes expressed in the ground tissues were characterized and validated by in situ hybridization and for one of them GATA6 inactivated by CrisprCas9 mediated targeted mutagenesis. Transcriptomics cluster were found coherent with the ATAC-seq analysis of the meristematic zone and the elongation zone of the radicle meristem. Finally a comparison with Arabidopsis RAM scRNA-seq data base revealed for some clusters, associated with epidermis, xylem and phloem differentiation, conserved core gene list but also specie specific genes that likely reflect the specificities and divergences between monocot and dicot.

I do not have the skills to assess the relevance of the analysis and inference methods used here but I find that this article is original and provides a sum of new data which will be invaluable for the study of plant development. For this reason I am in favor of its publication. However, before this is considered, I would like the two major following points to be taken into account by the authors.

Thanks for your positive comments and useful suggestions.

1. 194-197: the phenotype presented for the GATA6 KO mutant (supfig4) are only qualitative and visual. I should expect here some quantification and statistical analysis of the whole plant level phenotype and also a more precise description including quantification and statistical analysis of the phenotype observed at the histological level in the RAM.

Do you obtained only one KO allele and if yes do you think it is enough to support definitive conclusion? it should be better to have at least two different alleles. Do you have also transformed no mutated segregating line for control in addition to the wild type?

I think that the presentation and the analysis of the GATA6 phenotype should be strongly

improved to support the conclusion "these findings substantiate the precision of scRNA-seq in identification of novel rice root mutants at the cell-specific level." that constitutes a key point of this article.

Re: Thanks for your suggestions. We fully agree. To confirm the phenotype of *Osgata6* mutants, we studied the root phenotypes of another knock-out line (OsGATA6 KO #2) which was generously provided by Dr. Wen-Hui Lin (SJTU). The quantitative analysis of root lengths and RAM size clearly revealed an decreased meristematic activity in the *Osgata6* mutants.

Figure R3-1. The root phenotype of *Osgata6* mutants.

See also Supplementary Fig. 4.

2. In the conclusion it is claimed that this work may help identify key genes involved in plant adaptive development. As mentioned in the introduction the major part of the rice root system is

constituted by adventitious roots and the adaptive development of the rice root system is based on the modulation of adventitious and/or lateral root initiation or development. For this reason I think that in addition to compare the newly scRNA-seq data with similar dataset obtained in Arabidopsis, it should be interesting to know as far your obtained data overlap with exhaustive transcriptomic data obtained during initiation (Lavarenne et al., 2019, Plant Journal 100, 954-968) or in different zones and tissues (Takehisa et al., Plant J. 2012;69: 126–140) of rice adventitious roots. In particular in this last study there is a root cap specific gene list that could help to identify a root cap cluster in your data sets. In addition I think this comparison between embryonic and post embryonic rice root transcriptomes will be interesting for the scientific community.

Re: Thanks for your valuable suggestions.

1) Unfortunately, we noticed that the datasets generated by recommended two papers were based on microarray assays. Therefore, it is technically difficult to compare these datasets with our scRNA-seq dataset directly. Moreover, Takehisa et al. paper used a mixed sample comprising of EpiExo, Cortex and EndStele tissues from the R2R3 and R7 regions. Please note that the assignment of cell clusters in our dataset was confirmed not only by RNA *in situ* hybridization assays, but also by the promoter reporter analysis.

2) Root cap cell cluster. According to your suggestion, we examined the expression pattern of 653 root cap enriched genes identified by Takehisa et al. in our scRNA-seq dataset. Based on their expression pattern, we can classified these genes into three categories: 1) 193 genes belong to cluster-enriched genes (Figure R3-2a, b). However, these genes are widely distributed in all the cell clusters ; 2) 447 genes do not exhibit cluster-specific expression pattern (we showed 8 representative genes in Figure R3-2d); 3) the expression of the rest 13 genes cannot detected in our atlas. Taken together, the above results indicate that we could not faithfully identify a root cap cell clusters based on the 653 root cap enriched genes. It is very likely that the rice root cap cells are resistant to cell wall digestion. Interestingly, *Os03g0247200*, one of the 653 root cap enriched genes, was expressed in cluster 14 (root cap junction cell cluster; Figure R3-2c, see also Supplementary Fig. 10l).

Figure R3-2. The expression pattern of 653 root cap genes in root cell atlas.

3. Another important point 18, 76 and after : The rice primary root must be named radicle not radical.

Re: I am sorry for this confusion. It is a typo and has been fixed.

Two minor points:

4. Line 29: "... is therefore crucial for future crop design."

I don't think this is the most important perspective of this work to be highlighted. The most important newly perspective for me is relative to a better understanding of RAM functioning in particular in an evolutionary point of view between dicot and monocot

Re: Thanks. We have revised this sentence according to your suggestion. We wrote "Dissection of the developmental trajectories and the transcriptional networks that underlie them is therefore crucial for better understanding of the function of the root apical meristem in both dicots and monocots."

5. Line 190 "... is expressed [in the] center of the RAM...."

Re: Fixed.

Reviewer #4 (Remarks to the Author):

Unlike Arabidopsis, which the molecular basis of root development and setting up root cell identify is well studied; that in rice is still obscure. To understand the developmental trajectories and transcriptional networks in rice root, the authors conducted scRNA sequencing and chromatin accessibility survey of rice radicles. Further analysis showed that the rice root tip is composed of highly heterogeneous cells. Different clusters can be assigned to 10 major root cell types according to the scRNA-seq data. They further showed that this analysis could help to identify new players for root development in rice. Overall, the authors presented a novel rice root single cell transcriptome and revealed the conserved and divergent root developmental pathways between rice and Arabidopsis, which will be of great value for the field. However, there are a few concerns need to be clarified.

Thanks for your supportive comments and constructive suggestions.

1. First of all, the authors need to give a brief description of the quality of scRNA-seq data, for example, the median number of genes and transcripts detected per cell. Genes induced by protoplasting should be removed prior to analysis.

Re: Thanks for your suggestions.

1) The general information of scRNA-seq datasets including median number of genes and transcripts detected per cell are given in Supplementary Table 1.

2) To evaluate the effect of protoplast on scRNA-seq, we plotted the proportion of protoplasting genes (i.e. the genes whose expression levels are greatly affected by cell wall digestion during protoplasting) identified by the Qian lab (Supplementary Table 2; Wang et al., bioRxiv, 2020/<https://doi.org/10.1101/2020.01.30.926329>) in all the 21 cell clusters. As you can see in Figure R4-1 (Supplementary Fig. 1c), the proportion of protoplasting genes is below 1%, and none of cell clusters are enriched in these genes. Thus, this result clearly demonstrates that protoplasting procedure does not affect subsequent cell clustering.

Figure R4-1. Proportion of rice protoplast genes in each cell cluster.

Y-axis, cell cluster; X-axis, the proportion of rice protoplast genes within a cell cluster (%). see also Supplementary Fig. 1c.

2. Could the authors explain the absence of the well-known transcription factors for root development, like Scarecrow and PLT exc. Furthermore, it's quite strange that the authors could detect the expression of OsSHR1 and OsRHL1 in sclerenchyma. Both have been showed to be absent in these cells (Cui et al., 2007, Ding et al., 2009). This information needs to be carefully verified.

Re:

1) As shown in Figure R4-2 (see below; Supplementary text and Supplementary Fig. 13a), we now demonstrate the expression of *PLT*, *OsSHR* and *OsSCR* genes on the UMAP. *PLT* genes are mainly expressed in the meristematic cell clusters. In contrast, both *OsSCR1* and *OsSCR2* are mainly expressed in endodermis, whereas *OsSCR2* transcripts could be also detected in the cortex cell cluster. *OsSHR* genes are predominantly expressed in meristematic and endodermic cell clusters.

Figure R4-2. UMAP showing the expression pattern of *WOX5*, *PLT*, *SCR* and *SHR* genes.

Inset, the meristematic cell clusters. see also Supplementary Fig. 13a.

2) The scRNA-seq dataset unavoidably has some noise. As shown in Supplementary Fig. 9, *OsRHLL1* was mainly expressed in the epidermal cells. As such, it was used as a marker gene (a cluster-enriched gene) for the EC cluster. As shown in Figure R4-2 (Supplementary Fig. 13a), *OsSHR1* is highly expressed in the meristematic cell clusters. Along the pseudotime (Fig. 3i,j), both *OsRHLL1* and *OsSHR1* are expressed in early differentiation stage, and did not serve as marker genes (cluster-enriched genes) for sclerenchyma cells.

3. As hormones play important role in the root development, it might be better for the authors to analyze the pattern of hormone synthesis and responses in different cell clusters.

Re: Thanks for your suggestion. However, our knowledge of hormone biosynthesis and signaling pathways in rice is still fragmented. As such, we could not annotate all the orthologous genes involved in hormone biosynthesis and signaling transduction in rice. Nevertheless, in response to reviewer' request, we called some genes involved in auxin transport (PIN) and signaling (ARFs and AUX/IAAs) on the UMAP (see Supplementary text and Supplementary Fig. 13b).

Figure R4-3. Expression pattern of known genes involved in root development.

see also Supplementary Fig. 13b.

4. Because the authors did not detect QC cells in their scRNA-seq data, this information should be discussed. Furthermore, did they remove the dataset of QC cell cluster in the inter-species comparison analysis?

Re:

1) As shown in Figure R4-2 (see above; Supplementary text and Supplementary Fig. 13a), *OsWOX5* was expressed in meristematic cell clusters. However, due to low number of *WOX5*⁺ cells, we could not faithfully annotate a cell cluster corresponding to the QC in our dataset.

2) For inter-species comparison analysis, we include all the meristematic cells.

3) We now discuss the reason why we could not faithfully detect the QC cells in our datasets in the Supplementary text. We wrote “*WUSCHEL-RELATED HOMEODOMAIN BOX5 (WOX5)* is exclusively expressed in the Arabidopsis QC. *OsWOX5* was expressed in meristematic cell clusters (Supplementary Fig. 13a). However, due to low number of *WOX5* positive cells, we could not faithfully annotate a cell cluster corresponding to rice QC in our atlas.”.

5. In line 155-160, the description of the development of ground tissue is a bit miss leading.

Normally, the ground tissue in rice consists of exodermis, sclerenchyma, cortex and endodermis (Rebouillat et al., 2009). A detailed description of formative cell division pattern in rice ground tissue stem cells can be found in Ni et al., 2014 (Plant Biology). The authors need to revise the description.

Re: Fixed. We wrote “Successive periclinal divisions of the ground tissue initial cell generate exodermis, sclerenchyma, cortex layers and endodermis.”.

REVIEWERS' COMMENTS

Reviewer #1 (Remarks to the Author):

The authors properly answered my comments. I do not have additional suggestions.

Reviewer #2 (Remarks to the Author):

Dear authors & editor,

Thank you for the careful responses to my comments, and the corresponding revisions to your manuscript. The authors have addressed my concerns.

Reviewer #4 (Remarks to the Author):

All my concerns have been clarified. I'm happy with the current version